# A novel regional irrigation water productivity model coupling irrigation-drainage driven soil hydrology and salinity dynamics, and shallow groundwater movement in arid regions, China

Jingyuan Xue[1], Zailin Huo[1*], Shuai Wang[1], Chaozi Wang[1], Ian White[2], Isaya Kisekka[3], Zhuping Sheng[4], Guanhua Huang[1], Xu Xu[1]

[1]College of Water Resource and Civil Engineering, China Agricultural University, Beijing 100083, China.

[2]Fenner School of Environment & Society, Australian National University, Fenner Building 141 Canberra ACT 0200.

[3]Univerisity of California Davis, Department of Land, Air and Water Resources & Department of Biological and Agricultural Engineering

[4]Texas A&M University, Agriculture Research and Extension Center, El Paso, USA

[*] Correspondence to: Zailin Huo (huozl@cau.edu.cn)

## **Abstract:**

The temporal and spatial distribution of regional irrigation water productivity (RIWP) is crucial for making agricultural related decisions, especially in arid irrigated areas with complex cropping patterns. Thus, we developed a new RIWP model for an irrigated agricultural area with complex cropping patterns. The model couples the irrigation and drainage driven soil water and salinity dynamics and shallow groundwater movement, to quantify the temporal and spatial distributions of the target hydrological and biophysical variables. We divided the study area into $1 \text{ km} \times 1\text{km}$ hydrological response units (HRUs). In each HRU, we considered four land-use types: sunflower fields, wheat fields, maize fields and uncultivated lands (merely bare soil). And we coupled the regional soil hydrological processes and groundwater flow by taking a weighted average of the water exchange between unsaturated soil and groundwater under different land-use types. The RIWP model was calibrated and validated using eight years of hydrological variables obtained from regional observation sites in a typical arid irrigation area of North China, Hetao Irrigation District. The model reasonably well simulated soil moisture and salinity, as well as groundwater table depths and salinity. Overestimations of groundwater discharge were detected in calibration and validation due to the assumption of well-operated condition of drainage ditches, and regional evapotranspiration (ET) were reasonably estimated while ET in uncultivated area was slightly underestimated in RIWP model. Sensitivity analysis indicates that soil evaporation coefficient and specific yield are the key parameters for RIWP simulation. The results showed that, from 2006 to 2013, RIWP decreased from maize to sunflower to wheat. It was found that the maximum RIWP can be reached when groundwater table depth is in the range of 2 m to 4 m, regardless of irrigation water depths applied. This implies the importance of groundwater table control on RIWP. Overall, our distributed RIWP model can effectively simulate the temporal and spatial distribution of RIWP and provide critical water allocation suggestions for decision makers.

**Keywords**: Arid irrigated area, regional water productivity model, shallow groundwater, irrigation process, drainage, cropping patterns

## 1. Introduction

Under the increasing food demand of growing populations worldwide, water resources is limiting food production in many areas (Kijne et al., 2003; Fraiture and Wichelns, 2010). Especially, in arid and semi-arid regions of the world, where irrigated agriculture accounts for about 70 to 90% of the total water use (Jiang et al., 2015; Gao et al., 2017, Dubois, 2011), water deficit and related land salinity are the two major limitations to agricultural production (Williams, 1999; Xue et al., 2018). To maximize agricultural production, the improvement of irrigation water productivity (IWP) is vital (Bessembinder et al., 2005; Surendran et al., 2016). IWP is defined as the crop yield per cubic meter of irrigation water supplied, and the unit of IWP is kg/m$^3$ (Singh et al., 2004).

Furthermore, by changing hydrological processes, irrigation and drainage affect water and salt dynamics in crop root zone, groundwater, and, eventually, crop production (Morison et al., 2008; Bouman, 2007). Specifically, in arid region, irrigation-caused deep seepage is the mainly recharge of groundwater. Shallow groundwater can in turn go upward and contribute to crop water use by capillary action, which means the irrigation seepage can be reused by the crop growth to improve IWP. Thus, RIWP analysis requires the quantification of the complex agro-hydrological processes, including soil water and salt dynamics, groundwater movement, crop water use and crop production. Various methods have been used to evaluate IWP, such as field measurements (Talebnejad et al., 2015; Gowing et al., 2009), remote sensing (Zwart and Bastiaanssen, 2007), and distributed hydrological models (Singh, 2005; Jiang et al., 2015; Steduto et al., 2009). Field experiments have been widely used to evaluate the effect of water management on IWP (Talebnejad et al., 2015; Gowing et al., 2009), but field experiments are expensive and time consuming, making it unsuitable for regional evaluation of IWP. Conveniently revealing temporal and spatial distributions of ET and crop yields, remote sensing is commonly used to quantify regional IWP (Thenkabail and Prasad, 2008). However, remote sensing is looking at seeing the past IWP distribution, but cannot readily predict the impacts of water management practices on IWP.

Recently, distributed integrated crop and hydrologic models have been widely used to simulate the complex agro-hydrological processes coupled with salt dynamics and crop production (Aghdam et al., 2013; Noory et al., 2011; van Dam, 2008; Vanuytrecht et al., 2007). Taking advantages of

geographic information systems (GIS), distributed integrated crop and hydrologic models provide
precise simulations of regional hydrological processes and crop growth, by incorporating the
heterogeneity of soil moisture, salinity and texture, groundwater table depth and salinity, and
cropping patterns (Amor et al., 2002; Bastiaanssen et al., 2003a; Jiang et al., 2015; Nazarifar et al.,
2012; Xue et al., 2017).
There are two types of distributed hydrologic models that are used to monitor complex regional
hydrological processes: numerical distributed models, such as SWAT and MODFLOW, and
simplified distributed models, such as FARME (Kumar and Singh, 2003) and HEC-HMS (USACE,
1999) based on water balance equations. Numerical, process-based models consider the entire
complexity and heterogeneity of regional hydrological systems. MODFLOW is commonly used for
groundwater dynamics simulation (Kim et al., 2008). But it is limited in well-monitored large
irrigation areas, due to the large number of parameters and input data required. SWAT is used to
simulate land surface hydrologic and crop growth processes. It relies on the digital elevation model
(DEM) to delineate surface water flow pathways. However, many irrigation areas are quite flat, and
surface water flow pathways are controlled by irrigation and drainage systems, instead of terrain
elevation differences.
Simplified distributed models often employ mass balance equations to describe the soil water and
salt dynamics (Sharma, 1999; Sivapalan et al., 1996), which means less input parameters, and larger
spatial grids and temporal steps. However, the large spatial grids poorly reflect the regional complex
cropping pattern heterogeneity, and the large temporal steps cannot capture daily soil water and salt
dynamics which is essential for crop growth simulation. SWAT alone does not describe the complex
interactions between groundwater and soil water, which are fundamental in arid and semi-arid areas
with shallow groundwater.
After all, there are still two big challenges for developing a distributed integrated irrigation water
productivity models in irrigated areas. First, the networks of irrigation canals and drainage ditches
cause spatial heterogeneity in irrigation, drainage, deep percolation, canal seepage and groundwater
table depth within the irrigation area. But previous studies have overlooked the important role of
the networks of irrigation canals and drainage ditches in RIWP evaluations. Second, the multi-scale
matching problem comes out when coupling unsaturated and saturated zone in irrigation areas with

complex cropping patterns, as the spatial heterogeneity of cropping patterns is much stronger than

that of groundwater table depth. However, most of the existing distributed hydrological models

simulated the hydrological processes within the same hydrological response unit (HRU) between

unsaturated and saturated zones independently, but overlooked the lateral exchange of groundwater

between adjacent HRUs.

Therefore, the main objectives of our study are to (1) develop a RIWP model framework coupling

the irrigation and drainage processes, soil water and salt dynamics, crop water and salt response

processes, and lateral movement of groundwater and salt; and (2) analyze the distributed RIWP of

the study area and find the effects of crop type, irrigation water depth applied and groundwater table

depth on RIWP.

## 2. Methods

We will present a four-module integrated RIWP model, the coupling between the modules and one

case study evaluating the model performance.

### 2.1 Regional irrigation water productivity model

General descriptions will be given for the four modules and their integration, as well as the division

and connections of HRUs, and boundary conditions of the model. Then, detailed descriptions will

be given for each of the four modules: irrigation system module, drainage system module,

groundwater module, and field scale IWP module.

#### 2.1.1 General descriptions

A four-module integrated RIWP model was developed, to simulate the complex system including

water supply from irrigation open canals, field crop water consumption, groundwater drainage into

open ditches, and groundwater lateral flow.

**(1) Four modules and their integration**

The developed RIWP model couples an irrigation system module, a drainage system module, a

groundwater module and a field scale IWP evaluation module (Fig. 1). The irrigation system

module simulates the water flow along canals and the canal seepage to groundwater (the recharge

of the groundwater module), and it provides the amount of water available for field scale
irrigation. The drainage system module simulates the drainage to main drainage ditches from
groundwater, and this is the discharge of the groundwater module. The groundwater module is
used to simulate the groundwater lateral movement, the groundwater boundary for field scale
water-salt balance processes, and the groundwater level dynamics for the drainage module. In the
field scale IWP module, vertical movement of water and salt in soil profile is simulated, to obtain
the soil moisture and salinity of the crop root zone, and to calculate field scale irrigation water
productivity. This module provides deep percolation to the groundwater module and obtains
capillary rise to soil from the groundwater module. The above mentioned four modules will be
described comprehensively in 2.1.2 to 2.1.5.
**(2) Hydrological response units**
The irrigation area is spatially heterogeneous in terms of soil, land use, meteorology and
groundwater. To include the spatial heterogeneities in the simulation of regional water and salt
dynamics and its impact on crop growth, the irrigation district was divided into hydrological
response units (HRUs) (Kalcic et al., 2015). The HRU is an abstract artefact created by
hydrological developer and is like the smallest spatial unit of the model, which provides an efficient
way to discretize large watersheds where simulation at the field scale may not be computationally
feasible. In each HRU, soil texture and groundwater conditions are assumed to be homogeneous,
but different cropping patterns can exist. For example, sunflower fields, wheat fields, maize fields
and uncultivated lands. As the irrigation quota is different for different cropping patterns, the model
first runs field IWP model for each cropping pattern independently in each HRU, to obtain the soil
water and salt dynamics, IWP, and groundwater recharge. Then, the groundwater levels and salinity
of each HRU can be updated according to the area proportions of different cropping patterns in
each HRU. The groundwater flow is determined by pressure head gradient between adjacent HRUs.
**(3) Boundary conditions**
The upper boundary of the model is the atmospheric boundary layer above the plant canopy, which
determines reference ET, and precipitation. The main irrigation canals and drainage ditches directly
connect with groundwater and can be considered as the side boundaries in the model. With the
canal conveyance water loss deducted from the gross water supplied, the amount of water diverted
into the field can be calculated as the actual amount of irrigation. The local irrigation schedules of
different crops and the actual time of canal water supply are both considered to determine the actual
irrigation time and irrigation amounts. The lower boundary is the confining bed at the bottom of
phreatic layer. The phreatic layer is vitally important due to its vertical exchange with the
unsaturated soil zone in each HRU and its lateral exchange with adjacent HRUs to bond the whole
region together.

## 2.1.2 Irrigation system module

When irrigation water passes through canals, no matter lined or unlined, seepage loss occurs
which recharges groundwater. In a large irrigation area, there are many main, sub-main, lateral,
and field canals, which are categorized as the first-, second-, third-, and fourth-order canals,
respectively. During the water allocation period, canal seepage loss from different levels of
canals can be divided into two parts. One part is the seepage loss from the main and sub-main
canals, which are permanently filled with water and recharge directly into groundwater along the
route. The other part is the seepage loss from lateral and field canals, which are intermittently
filled with water and only recharge the groundwater units within their control area. Each HRU
has its corresponding groundwater unit, which is used when calculating lateral exchange of
groundwater between adjacent HRUs.
We calculated the decreasing water flow along canal, and water losses in main and sub-main canals
as follows (Men 2000):

$$\sigma = \frac{A}{100Q^m} \tag{1}$$

$$\sigma = \frac{dQ}{Qdl} \tag{2}$$

where $\sigma$ represents the water loss coefficient per unit length per unit flow in canal (m$^{-1}$). $A$ is the
soil permeability coefficient of canal bed (m$^{3m-1}$day$^{-m}$), m is the soil permeability exponent of canal
bed (-), and their values depend on the soil type of the canal bed (please refer to Guo (1997) for
the values). $Q$ represents the daily net flow in canal (m$^3$day$^{-1}$), and $dQ$ represents the daily flow
loss of the water conveyance within $dl$ distance in canal (m$^3$day$^{-1}$).
Thus, Eq. (1) is equal to Eq. (2), and they can be transformed into:
$$Q^{m-1}dQ = Adl \tag{3}$$

Integrations of both sides of Eq. (3) gives:
$$\int_{Q_L}^{Q_g} Q^{m-1} dQ = \int_0^L A\, dl \tag{4}$$

$$Q_L = (Q_g{}^m - ALm)^{1/m} \tag{5}$$

where $Q_g$ is the daily gross flow in the head of canal (m³day⁻¹), and $Q_L$ is the daily net flow in
canal at $L$ distance away from canal head (m³day⁻¹). Thus, flow loss in water conveyance process
can be calculated as follows:
$$Q_{Ls} = \frac{A}{100}(Q_g{}^m - ALm)^{(1-m)/m} \tag{6}$$

$$W_{ls} = Q_{ls}/(n_1 \times A_{su}) \tag{7}$$

where $Q_{Ls}$ is the daily groundwater recharge due to water conveyance loss in main and sub-main
canals (m³day⁻¹), $W_{ls}$ is the daily groundwater recharge per unit area due to water conveyance loss
in main and sub-main canals (mday⁻¹). $n$ represents the total number of HRUs along selected main
and sub-main canals (-), and $A_{HRU}$ is the area of each HRU (m²).
Lateral and field canals are densely distributed in the irrigated area, and they are intermittently
filled with low water flow. Thus, it is assumed that seepage from these canals uniformly
recharges groundwater units within their control area. The canal seepage is estimated by an
empirical formula:
$$W_{as} = I_n * \eta_{mc} * (1 - \eta_{sbmc}) + I_n * \eta_{mc} * \eta_{sbmc} * (1 - \eta_{lc}) + I_n * \eta_{mc} * \eta_{sbmc} * \eta_{lc} * (1 -$$

$$\eta_{fc}) \tag{8}$$

where $W_{as}$ represents daily groundwater recharge per unit area due to water conveyance loss in
lateral and field canals (mday⁻¹), and $I_n$ is daily irrigation water depth applied per unit area (mday⁻
¹). $\eta_{mc}$, $\eta_{sbmc}$, $\eta_{lc}$ and $\eta_{fc}$ are the utilization coefficient of main, sub-main, lateral and field canals,
respectively (-).
**2.1.3 Drainage system module**
In the drainage system module, only the groundwater draining into ditches is considered. Because
the precipitation directly on ditches is negligible in arid and semi-arid area. The drainage processes
are simulated based on the spatial distributions of main, sub-main, and lateral ditches, which are

grouped into the first-, second-, and third-order ditches, respectively. Drainage is estimated by comparing local groundwater levels and ditch bottom elevation. According to Tang et al. (2007), the groundwater drainage was calculated by:

$$D_g = \begin{cases} \gamma_d \times (h_{db} - h_g) & ; \ h_{db} > h_g \\ 0 & ; \ h_{db} < h_g \end{cases} \qquad (9)$$

where $D_g$ is daily groundwater drainage per unit area (mday$^{-1}$). $\gamma_d$ is drainage coefficient (-), which describes the groundwater table decline caused by the elevation difference between groundwater table and the streambed of the drainage ditch. And it depends on the underlying soil conductivity and the average distance between the drainage ditches. $h_g$ represents the daily groundwater table depth (mday$^{-1}$), and $h_{db}$ is the daily streambed depth of drainage ditch (mday$^{-1}$).

**2.1.4 Groundwater module**

For a plain irrigation area, usually groundwater levels are relatively flat on a large scale. In our model, it is assumed that groundwater lateral flow exists between one HRU and its four adjacent HRUs (Fig. 2). Using water table gradient, groundwater flow between current HRU and its adjacent HRUs can be calculated by:

$$W_{gr} = (K \times h \times B \frac{L_{ga} - L_g}{D}) / B^2 \qquad (10)$$

where $W_{gr}$ is the daily groundwater inflow of the current HRU from adjacent HRUs (mday$^{-1}$), and $K$ is the daily permeability coefficient of unconfined aquifers in the current HRU (mday$^{-1}$). $h$ represents the thickness of unconfined aquifers, which is the difference between water table and upper confined bed and varies with water table changes (m). $B$ is the length of groundwater unit (m) and here the value is 1km. $L_{ga}$ and $L_g$ represents the water table level of adjacent HRUs and the current HRU, respectively (m). $D$ is the distance between the center of the current HRU and the centers of its adjacent HRUs (m). There are three types of groundwater boundary conditions: river head (when the boundary HRU including irrigation canal and the daily river flux equals to the daily canal flux), river flux (when the boundary HRU including drainage ditches and the water heads in ditches are assumed constant and equal to the river head) and constant flux (when the boundary HRU is mainly barren area and no irrigation is applied, thus in our study 0 flux is assumed).

Based on the field scale simulation, groundwater lateral exchange, canal seepage and groundwater
drainage are added in the daily water and salt balance calculations of each groundwater unit at
regional scale:
$hg_i = hg_{i-1} - (1/S_y)(Pwg_{i-1} - Gwg_{i-1} - ext_{i-1} + W_{grup i-1} + W_{grdown i-1} + W_{grleft i-1} +$
$\qquad\qquad W_{grright i-1} + W_{ls i-1} + W_{as i-1} - D_{g i-1})$ (11)
$SCa_i = Za \times Sa_{i-1} + W_{grup i-1} \times Sa_{up i-1} + W_{grdown i-1} \times Sa_{down i-1} + W_{grleft i-1} \times$
$Sa_{left i-1} + W_{grright i-1} \times Sa_{right i-1} + (W_{ls i-1} + W_{as i-1}) \times Is_{i-1} - D_{g i-1} \times Sa_{i-1} +$
$\qquad\qquad\qquad\qquad\qquad Psg_{i-1} - Gsg_{i-1}$ (12)
where $W_{grup}$, $W_{grdown}$, $W_{grleft}$ and $W_{grright}$ are the daily groundwater lateral runoff per unit area into
the current groundwater unit from up and down or left and right adjacent groundwater unit,
respectively (mday$^{-1}$). $SCa$ is the daily soluble salt content in the saturated zone below the
transmission soil profile (mg m$^{-2}$day$^{-1}$). $Z_a$ is the thickness of the saturated zone which is the
difference between the groundwater table depth and the depth that groundwater table fluctuations
largely cannot reach (m). $Z_a$ only affect the soluble salt concentration in the groundwater salt balance,
while it has no effect on the water balance and groundwater fluctuation simulation. $Sa$, $Sa_{up}$, $Sa_{down}$,
$Sa_{left}$ and $Sa_{right}$ is the salt concentration of the current groundwater unit and its up and down or left
and right adjacent groundwater units, respectively (mg m$^{-3}$). $Is$ is the salt concentration of the
irrigation water (mg m$^{-3}$). $S_y$ represents the specific yield (-), which is the ratio of the volume of
water that can be drained by gravity to the total volume of the saturated soil/aquifer. $ext$ is the daily
groundwater extraction per unit area (mday$^{-1}$). $P_{wg}$ is the daily percolation water depth to
groundwater from the potential root zone (mday$^{-1}$), and $G_{wg}$ is the daily water depth supplied to the
potential root zone from shallow groundwater due to the rising capillary action (mday$^{-1}$). $P_{sg}$ and
$G_{sg}$ are the quantity of soluble salt in $P_{wg}$ and $G_{wg}$, respectively (mg m$^{-2}$day$^{-1}$). The detailed
calculations of the water and salt exchange components between unsaturated soil and groundwater,
such as $P_{wg}$ and $G_{wg}$, were described in our previously developed water productivity model at field
scale (Xue et al., 2018).
**2.1.5 Field scale irrigation water productivity module**
Cropping patterns are complex for each HRU and sometimes HRUs include uncultivated land, forest

land and other non-agricultural land. In our model, with high resolution land use map, different

cropping patterns can be separated to simulate soil water and salt processes, and the responses of

ET and crop yields to water and salt content of root zone. Here, we employed our previously

developed field IWP model to simulate field water, salt, ET and crop yield under shallow

groundwater condition (Xue et al., 2018). The soil profile is vertically divided into four soil zones:

the current root zone, the potential root zone, the transmission zone, and the saturated zone. In each

HRU, the soil water and salt balance processes, and water productivity are independently simulated

for each cropping pattern under its corresponding groundwater unit condition. For uncultivated

lands, only water and salt balance are simulated, and its IWP is 0. Then, the water and salt exchange

between unsaturated soil and groundwater of different cropping patterns are weighted averaged by

area proportion. Finally, the weighted averages are used to update daily groundwater table and

salinity (Fig. 3).

## 2.2 Modules coupling and calculating flowchart

The simulation was by daily temporal step and by HRU spatial step. The irrigation system module

simulates the canal seepage to groundwater and the field irrigation water amount. And the canal

seepage to groundwater is the recharge of the groundwater module, while the field irrigation water

amount is the input of the field IWP module. The drainage system module simulates the

groundwater drainage to drainage ditches, which is the discharge of the groundwater module. The

groundwater module is used to simulate the groundwater table depth, which is the input of the field

IWP module and also the input of the drainage module. In the field scale IWP module, the deep

percolation to groundwater under different cropping patterns are simulated independently and their

weighted average is the recharge of the groundwater module. The salt exchange is simulated

together with water exchange. The groundwater module is used to simulate the groundwater lateral

movement between the current HRU and its adjacent HRUs to update the groundwater level at next

time step. By coupling the irrigation system module, drainage system module and groundwater

module with the field IWP model, this RIWP model simulates the temporal and spatial distribution

of IWP in the whole irrigation area from the beginning to the end of the growing season.

The model was implemented in a combination of ArcGIS, MATLAB, and Microsoft Excel (Fig. 4).

The HRUs was created in ArcGIS as fishnet, with each grid numbered. In MATLAB, the HRUs
were represented by a matrix and the daily time step was represented by a vector. At each time step,
all the HRUs were traversed by a nested loop. Then the updated information for the current time
step was used to calculate the next time step.   Microsoft Excel stored ArcGIS vector layer and its
attribute data for MATLAB modeling, and also stored MATLAB output results for ArcGIS analysis
and visualization.
Considering the high spatial heterogeneity, meteorological data need to be collected from all the
weather stations within or close to the study area. Distribution of soil physical properties, moisture
and salinity in unsaturated soil, groundwater table depth and salinity, need to be collected from
many observation sites, which are uniformly or randomly spread over the study area. Then, each
data set can be interpolated in ArcGIS by inverse distance weight to obtain a spatial distribution
vector layer. For each layer, the average value in each HRU are calculated by ArcGIS using
geometric division statistics. The vector layer of irrigation control zones and the vector layer of
drainage control zones is respectively overlaid with the HRU division layer in ArcGIS, to obtain the
HRU numbers controlled by each irrigation control zone and each drainage control zone. The HRU
numbers controlled by the same zone are stored in the same matrix for batch simulation in MATLAB.
In MATLAB, soil water and salt balances and field scale IWP for main crops are simulated
simultaneously for each HRU; whereas, groundwater lateral exchange are simulated between
adjacent HRUs. At the end of the model simulation, soil moisture and salinity, groundwater table
depth and salinity, ET, crop yield and IWP for different land use types in each HRU can be obtained.
Then, the area proportion weighted average in each HRU can be imported into ArcGIS to visualize
the spatial distribution.
**2.3 Model evaluation**
We will provide a case study using the above developed new RIWP model, to test its applicability,
and to provide sensitivity analysis of the parameters.
**2.3.1 Description of study area and data**
As a typical sub-district of the Hetao Irrigation District, the Jiefangzha Irrigation District (JFID) is

a typical arid irrigated area with shallow groundwater, resulted from its arid-continental climate, over years of flood irrigation, and poor drainage systems (Fig. 5). Located in the Hetao Plain, the JFID is very flat with an average slope of 0.02% from southeast to northwest (Xu et al., 2011). The mean annual precipitation is only 155 mm, of which 70% occurs between July to September; while the mean annual potential evaporation is 1938 mm. The mean annual temperature is 7°C, with the lowest and highest monthly average being −10.1°C and 23.8°C in January and July, respectively. The JFID covers an area of 0.22 Mha, of which 66% is irrigated farmland area. Wheat, maize and sunflower as the main crops in this region, taking up more than 90% of the irrigated farmland area. The $12\times10^8$ m$^3$ annual irrigation water is diverted from the Yellow River. Due to the poor maintenance of drainage ditches, it is quite common in this area to have poor drainage situations. Therefore, the annual average groundwater table depth ranges from 1.5 to 3.0 m during the crop growing season. Soils in the JFID are spatially heterogeneous and primarily composed of silt loam in the northern region and sandy loam in the southern region. Shallow groundwater table and strong evaporation makes soil salinization a very serious problem in this area, which is becoming the main constraint of crop production.

An irrigation and drainage network include four main irrigation canals, sixteen sub-main irrigation canals, five main drainage ditches, and twelve sub-main drainage ditches are controlling the water movement in the JFID (Fig. 5). The streambed depths of the regional main, sub-main and lateral ditches were collected by a regional survey in 2016. Daily water flow data in the main and sub-main irrigation canals and monthly data of the five main drainage ditches were obtained from the local Irrigation Administration Bureau. A total of 55 groundwater observation wells are installed in the JFID (Fig. 5). Groundwater level was measured on the 1[st], 6[th], 11[th], 16[th], 21[th] and 26[th] of each month, and groundwater salinity was measured 3 times each month. Near the groundwater observation wells, soil moisture was measured four times, and soil electrical conductivity was measured once before wheat sowing and once before autumn irrigation. Due to the spatially homogeneous climate in JFID, daily meteorological data (air temperature, humidity, wind speed and precipitation) was obtained from Hangjinghouqi weather station for the calculation of regional reference ET.

HJ-1A, HJ-1B and Landsat NDVI images with 30 m resolution during the period of 2006-2013 were downloaded from the official website of China Centre for Resources Satellite Data and Application

(2013) and USGS (2013), to determine the annual cropping pattern distributions. Due to the lack of measured ET, the ET estimated by SEBAL model using MODIS images from NASA (2013) was used as a reference to compare with simulated ET values (Bastiaanssen et al., 2003b).

**2.3.2 Parameterization of distributed RIWP model**

The JFID was divided into 2485 1km×1km HRUs (Fig. S1a in the supplementary material). In terms of boundary conditions, the upper Quaternary 4 aquifer layer was regarded as the phreatic layer in the model. It was modeled as an aquitard with loamy soil. From north to south, the thickness of aquifer in JFID varies from 2 to 20m with an average of 7.4m (Bai et al., 2008). Thus, the initial value of the average thickness of unconfined aquifer is set as 7.4m. The water level contour maps of JFID during 1997-2002 by Bai (200) were used to determine the direction of water flow near the groundwater boundary. Based on the topography conditions, land-use types, locations of main canals and ditches, and directions of water flow, the regional phreatic layer was divided into 5 zones with river, drainage and impervious boundary conditions (Fig. S1b).

The JFID was divided into four irrigation control sections and five drainage control sections, each section was controlled by one main irrigation canal or one main drainage ditch. These sections were further divided into 48 irrigation control sub-areas and 17 drainage control sub-areas, each sub-area was controlled by one sub-main irrigation canal or one sub-main drainage ditch (Fig. S2). The sunflower fields, wheat fields, maize fields and uncultivated lands are the four cropping patterns, i.e., land-use types, in the RIWP model. In many other researches about distributed hydrological models, when considering the applied irrigation schedule the sowing and irrigations of a particular crop were just set as on the same day over the whole study area, which may be a simplification of actual conditions (Singh, 2005). In our study, the irrigation time and irrigation water amount of each HRU were co-determined by both the local irrigation schedule of the three main crops, and the actual water amount flowing into the fields.

The simulation period was from April $1^{st}$ to September $20^{th}$, which covers the growing seasons of all the three main crops. The initial crop parameters were set as the default values suggested for sunflower, wheat, and maize by Allen et al. (1998). The empirical values of regional canal utilization and ditch drainage coefficient were obtained from Jiefangzha administration.

**2.3.3 Model calibration and validation**

To comprehensively evaluate the accuracy and reliability of the model, the data in years 2010-2013 and in years 2006-2009 was respectively used as calibration and validation dataset. The daily measured soil moisture content of crop root zone ($\theta$), electrical conductivity of soil water (EC), groundwater table depth ($h_g$) and groundwater salinity, were calibrated with measured data from the 22 soil water and salt observation sites and 55 groundwater observation sites (Fig. 5), which were mentioned in section 2.3.1. The RIWP simulated regional ET for each HRU was calibrated by the remote sensing based ET images obtained once per 8 days. The regional drainage processes was calibrated by the monthly groundwater drainage data from main ditches, in which the simulated drainage of each main ditch was the sum of drainage of its controlling HRUs. Overall, the soil hydraulic parameters, the crop water productivity related coefficient, and the canal conveyance and ditch drainage parameters were all calibrated with observed data in years 2010-2013, and then validated with observed data in years 2006-2009.

To quantify the model performance, the root mean square error (RMSE), the Nash and Sutcliffe model efficiency (NSE) and the coefficient of determination ($R^2$) were used as the indicators. RMSE was used to measure the deviation of simulated values from the measured ones, NSE was commonly used to verify the credibility of the hydrological model, and $R^2$ represented the degree of linear correlation. The indicators were calculated as follows:

$$RMSE = \left[\frac{\sum_{i=1}^{n}(Output_s - Output_o)^2}{n}\right]^{0.5} \tag{13}$$

$$NSE = 1 - \frac{\sum_{i=1}^{n}(Output_s - Output_o)^2}{\sum_{i=1}^{n}(Output_o - Output_m)^2} \tag{14}$$

$$R^2 = 1 - \frac{\sum_{i=1}^{n}(Output_o - \overline{Output_o})(Output_s - \overline{Output_s})}{\sqrt{\sum_{i=1}^{n}(Output_o - \overline{Output_o})^2}\sqrt{\sum_{i=1}^{n}(Output_s - \overline{Output_s})^2}} \tag{15}$$

where *n* is the number of simulations; *Output_s* and *Output_o* are simulated and observed values of model outputs, respectively; $\overline{Output_s}$ and $\overline{Output_o}$ are the average values of simulated and observed model outputs, respectively. The *RMSE* indicates a perfect match between observation and simulation when it equals 0, and increasing *RMSE* values indicate an increasingly poor match. Singh et al. (2005) stated that *RMSE* values less than 50% of the standard deviation of the

observed data could be considered low enough as an indicator of a good model prediction.
Ranging between $-\infty$ and 1, the NSE indicates a perfect match between observed and predicted
values when it equals to 1. Values between 0 and 1 are generally considered as acceptable levels
of performance, whereas values less than 0.0 indicate that the simulation is worse than taking an
average of observation, which indicates unacceptable performance. The $R^2$ ranging between 0 and
1 describes the proportion of the variance in the observed data, in which higher values indicating
less error variance. Typically, $R^2 > 0.5$ is considered acceptable (Santhi et al., 2001).

**2.3.4 Global sensitivity analysis**

To find the key parameters significantly impacting the model output, a global sensitivity analysis
was conducted. The analysis related the changes in three output variables—RIWP, groundwater
table depth and groundwater salinity—to eight parameters in the RIWP model. The Latin Hypercube
Sampling (LHS) (please see Mckay, 1979; Muleta et al., 2005; Wang et al., 2008 for detailed
descriptions of the sampling method), a typical sampling method for sensitivity and uncertainty
analysis, was used to sample the parameter space. According to Dai (2011), to ensure that the test
points were evenly distributed in space and to guarantee the accuracy of the test, the test number
was set as 20, more than double of the parameter number which was 8. For uniform distributions,
the parameter range was subdivided into 20 equal intervals. Each interval was sampled only once to
generate random values of the possible parameter sets. The possible parameter value ranges referred
to the local measurements, survey data and relevant research papers. Additionally, considering the
spatial heterogeneity of the three output variables, 22 evenly distributed groundwater observation
sites in JFID were selected for the global sensitivity analysis. Based on the LHS method, 20 groups
of parameter combinations were obtained and the simulation was run for 20 times. Finally, the
sensitivity of the three output variables to the eight parameters were determined in SPSS Statistics.
The absolute values of the obtained Standardized Regression Coefficients (SRCs) quantified the
significance of each parameter to each output variable (Table 1) (Cheng et al., 2018; Cannavó,
2012). And the plus or minus sign of the SRCs indicated the positive or negative correlations
between the corresponding parameter and output variable pairs.

## 3. Results and Discussion

### 3.1 Model performance

Good agreements were obtained by RIWP model in simulating IWP and hydrological components during the calibration and validation periods. Table 2 tabulated the calibrated parameters describing crop growth and water usage, and Table 3 tabulated the possible variation ranges and calibrated values of the parameters describing soil hydraulic characteristics and irrigation and drainage system. The agreement between the observed and simulated soil moisture content in crop root zone both in calibration (Fig. 6a, $RMSE$=2.867 cm$^3$ cm$^{-3}$, $NSE$=0.330, $R^2$=0.502) and validation (Fig. 6b, $RMSE$=2.989 cm$^3$ cm$^{-3}$, $NSE$=0.232, $R^2$=0.548) indicates the reasonable performance of the RIWP model. The good performance of the RIWP model was also indicated by the simulation of the soil salt content both in calibration (Fig. 6c, $RMSE$=1.108 dS m$^{-1}$, $NSE$=0.612, $R^2$=0.657) and validation (Fig. 6d, $RMSE$=1.205 dS m$^{-1}$, $NSE$=0.525, $R^2$=0.590). The simulated and observed groundwater table depth (Fig. 6e, $RMSE$=0.786m, $NSE$=0.424 and $R^2$=0.509 in calibration; Fig. 6f, $RMSE$=0.667m, $NSE$=0.637 and $R^2$=0.504 in validation) and groundwater salinity (Fig. 6g, $RMSE$<10%, $NSE$=0.813 and $R^2$=0.815 in calibration; Fig. 6h, $RMSE$<10%, $NSE$=0.604 and $R^2$=0.730 in validation) at 55 observation sites are in good agreement as well.

The model did not perform very well on simulating groundwater drainage. The overestimated drainage (Fig. 6i-j) was due to the different operating conditions of the drainage ditches of the different order. Remember that we classified the main, sub-main and lateral drainage ditches into the first-, second- and third-order ditches, respectively. In the model, for each year, we adopt same drainage coefficient for all the ditches of the different orders, assuming a well operated condition. However, the actual operating conditions of the ditches of the different orders cannot be the same, resulting in the simulation discrepancy.

The ET simulated by the RIWP model (ET$_{IWP}$) and the ET estimated by the SEBAL model using MODIS images (ET$_{RS}$) agrees well both in calibration (RMSE=1.918mm, $NSE$=0.274 and $R^2$ = 0.561) and in validation (RMSE=2.132mm, $NSE$ =0.189 and $R^2$ =0.498) (Fig. 6l). Furthermore, the comparison of the spatial distribution of cumulative ET$_{IWP}$ and ET$_{RS}$ during crop growth season showed that ET$_{IWP}$ was lower than ET$_{RS}$ in uncultivated area, while they agreed well in farmland (Fig. S3). The uncultivated area, merely bare soil, accounted for about 34% of the JFID, and the

$ET_{IWP}$ of uncultivated area was merely soil evaporation. This, resulted in the underestimation of
actual ET in uncultivated area compared to the ET acquired by remote sensing images, which was
consistent with previous studies (Singh, 2005; Tian et al., 2015). Besides, the cumulative $ET_{RS}$ was
taken by the 8 times of daily ET on satellite acquisition date, thus using the non-representative $ET_{RS}$
above the average daily value may also result in the underestimation of $ET_{IWP}$.
To test the model performances under different cropping patterns, one representative site was
selected for each cropping pattern to compare the observed and simulated time series of groundwater
table depth (Fig.7). Results indicated that the model can adequately capture the groundwater
dynamics at the four representative sites. Occasionally, the simulated groundwater table depth
declines fast, while the observed value rises. This is most likely due to the fact that we ignored the
time lag between groundwater recharge from soil and deep percolation. In the uncultivated area
(Fig.7a), simulated groundwater table level presented a slower and more flat decreasing trend than
measured value. By assuming a completely non-vegetation coverage condition of uncultivated area
while it is not actually the case, estimated groundwater evapotranspiration driven by capillarity will
become smaller than its actual value, in which small vegetation will transpires amounts of water
from soil and soil moisture is relatively low thus groundwater evapotranspiration is higher.
**3.2 Global sensitivity analysis**
Recall that the global sensitivity analysis was to determine the sensitivity of the three output
variables to eight parameters. The three output variables were RIWP, groundwater table depth, and
groundwater salinity; while, the eight parameters were those parameters describing soil hydraulic
characteristics and irrigation and drainage system, tabulated in Table 3. Specific yield ($S_y$), followed
by soil evaporation coefficient ($K_e$), are the two key parameters influencing the RIWP (Fig. 8a). The
specific yield indicated the readily available soil moisture released to crop root zone from shallow
aquifer under capillary action for crop consumption. Thus, its significant positive influence on
RIWP was explained. The soil evaporation coefficient indicated the proportion of water that
transferred into the atmosphere but was not used by crops. Therefore, its significant negative impact
on RIWP was expected. We concluded that for shallow groundwater buried area like JFID,
sometimes the effect of groundwater contribution on IWP would be greater than that of irrigation
water depth applied. Applying lots of shallow irrigation to the crops may reduce the deep percolation
and decrease the non-beneficial water use in evaporation. Applying fewer and deeper irrigation
water applied will result in deeper percolation meanwhile greater groundwater contribution to
beneficial crop water use. Thus, compared with lots of shallow irrigation applied, applying fewer
deeper irrigation schedule may have greater affect on IWP in arid regions with shallow groundwater.
And for both groundwater table depth (Fig. 8b) and groundwater salinity (Fig. 8c), specific yield
was the only key parameter. Canal seepage was expected to cause the variation of groundwater table
depth around the canal at the local scale. However, the results indicated that the variation of
groundwater table depth would be more susceptible to the local groundwater properties, i.e., specific
yield, than to canal seepage at the regional scale. We speculate that the lateral groundwater
movement might compensate the variation of groundwater table depth caused by the canal seepage.
Salt moves with water. Thus, the variation of groundwater salinity was also dominated by the
specific yield. Due to the high sensitivity of IWP, groundwater table depth and salinity to the specific
yield, it is highly recommended to use spatially variable values of specific yield rather than a
constant one as a model input if it is available, which could greatly enhance the evaluation accuracy
of the RIWP model. Also, it is indicated that the permeability coefficient of unconfined aquifers ($K$)
did not significantly affect the IWP, groundwater table depth and salinity. Due to the lack of
measurement data in our study, we adopted a unified $K$ value for the whole study area, which also
make the model simulations reasonable for their insensitive to this parameter.
**3.3 Regional irrigation water productivity**
**3.3.1 Spatial distribution of irrigation water productivity**
Validated by the measured soil moisture and salinity, groundwater table depth and salinity, drainage
water depth and ET, especially, the year 2006-2013 time series of groundwater table depth under
the four cropping patterns, the developed RIWP model can be used to estimate the spatial
distribution of IWP for the three main crops over the period of 2006-2013 (Fig. 9). Note that these
IWP values were based on the simulated water balance and crop yields of individual HRU, which
may deviate to a certain extent from the real values. It can still represent the utilization of water

resources at the regional scale. We could see there are "red HRUs" in Figure 9 changing with time

and space due to different irrigation water depth applied under different groundwater conditions.

Even different crop species can result in big difference in IWP. As we mentioned before, the spatial

distribution of these three crops is very complex in JFID and field plot is small, thus we use remote

sensing data to obtain cropping pattern map with resolution of 30m*30m. Every HRU has these

three crops, thus we can simulate IWP for each main crop in every HRU. The RIWP of the three

main crops showed a trend of decline during the period of 2006-2010 (Fig. 9a-e).This was mainly

attributed to the increasing irrigation quota, as the excess water lowered the IWP. Whereas, during

the period of 2011-2013 (Fig. 9f-h), the RIWP of the three main crops showed an increasing trend.

This was because that the irrigation quota was reduced over this period, and the contribution of

groundwater compensated the crop yield losses. With less irrigation water applied, the number of

"red HRUs" will increase along with it.

Under a given irrigation water distribution, the spatial distribution of ET was the key factor

controlling the RIWP distribution. And the spatial distribution of ET was fundamentally determined

by the solar energy, and the water and salt dynamics of soil. Recall that the climate and, therefore,

the solar energy, was homogeneous in JFID. Then, the spatial heterogeneity of RIWP must be

attributed to the water and salt heterogeneity caused by the spatial heterogeneity of the cropping

pattern, groundwater table depth, and irrigation and drainage networks. Particularly, when the

farmlands had limited supply of irrigation water, the groundwater table depth and salinity played an

important role on IWP. Through the drainage ditches, groundwater could drain both water and salt

out of the field, thus the groundwater table level declines and the soluble salt content going upward

along with groundwater evapotranspiration to crop root zone decreases. Despite the negative effect

of draining water on IWP, the positive effect of draining salt out of the field will positively affect

IWP. As we can see in Fig. 9, the simulated IWP values for three crops were lower in the south, west,

north and north-west of the JFID than in the other regions. The south of the JFID is the main canal

for water diversion, which provide higher irrigation quota than other regions, in which results in a

lower IWP. For the west of JFID, it is mainly uncultivated area, thus the IWP is lower than other

regions. In the north-west of the JFID, main drainage ditch received the drainage water with high

saline content from four sub-main ditches and drained all the way to the north of JFID. Ditch seepage

water with high salinity resulted in the severe soil salinization in the north and north-west of JFID, which will restrict the crop growth and lower the IWP. Thus, properly groundwater drainage management and dealing with salt accumulation at the end of main drainage ditches in an irrigated area is also a pressing and unsolved problem for increasing the "red HRUs", which needs to be figured out by irrigation managers.

As the major food-producing region of China, improving water productivity means producing greater amounts of food crops with less amount of water, based on local or regional potential. With declining access to water resources, farmers will need to grow different crops to maintain or increase crop production profitability in the future. The comparison between the RIWP of different crops (comparing the three columns in Fig. 9) showed that maize had the highest IWP, wheat had the lowest IWP, and the IWP of sunflower was in the middle. Therefore, modestly increasing the planting area of maize will improve the crop production per unit irrigation water amount. In addition, the RIWP of sunflower is a little higher than that of wheat, and the benefit and the salt tolerance of sunflower are both much higher than those of wheat. Thus, planting sunflowers should be promoted in the JFID when available irrigation water resources is declining in the future, and this practice will definitely increase the "red HRUs".

## 3.2.2 The impact of irrigation water depth applied and groundwater table depth on irrigation water productivity

In arid shallow groundwater area, irrigation water productivity (IWP) is affected by irrigation water depth (IWD) applied and groundwater table depth ($h_g$). In all the four simulated $h_g$ ranges, IWP decreased when IWD increased (Fig. 10a), which was consistent with Huang et al. (2005). Moreover, the magnitude of IWP decrease per unit increase of IWD was different under different $h_g$ ranges. The magnitude of IWP decrease under shallower $h_g$ was smaller than that under deeper $h_g$. This effect of increasing $h_g$ on the relationship between IWP and IWD was consistent with Gao et al. (2017). The above results indicate that when irrigation water is insufficient, groundwater can compensate the crop water demand. However, when irrigation water is excessive, a large proportion will eventually drain through the drainage ditches, and the IWP drops. Additionally, among the four $h_g$ ranges, the highest IWP was obtained in the range of 2-3m (Fig. 10b), which

was consistent with Xue et al. (2018). This indicates that a $h_g$ deeper than that provides insufficient water for crop growth; whereas, a $h_g$ shallower than that will increase root zone soil salinity and salt stress of crops. The negative effect of shallow groundwater salinity can also be found in Fig. 10a when $h_g$ is less than 2m, and it indicates that when irrigation applied decreased from 300<IWD<400mm to 200<IWD<300mm it leads to decreases in IWP, which is caused by faster reduction of ET than irrigation applied. Shallow buried groundwater contribution will make up for ET reduction when smaller irrigation water applied, thus there exists another reason accelerate the reduction of ET. We deduced that less irrigation water will weaken the role of irrigation on salt leaching and result in more severe salinization in crop root zone. The negative effect of salt stress on crop water use is greater than the positive effect of shallow groundwater contribution on crop water use at this situation. Thus, keeping the groundwater table depth in the optimal range and sustainable is of great importance to reach higher crop IWP at the regional scale, irrigation managers may need to reasonably determine the irrigation quota and constantly maintain the drainage system. Groundwater sustainability includes spacing withdrawals to avoid excessive depletion and taking measures to safeguard or improve groundwater quality. To achieve this, regional irrigation managers may need to take monitoring efforts to establish historic and current conditions, research to model groundwater systems, forecast future variation, and policy to manage activities influencing groundwater table and quality.

## 4. Conclusions

In view of the heterogeneous conditions of irrigated areas, taking fully consideration of the supply, consumption and drainage processes of irrigation water and groundwater, a distributed RIWP model was developed to couple the irrigation water flow processes along main canals and drainage processes, water and salt transport processes in soil profile, groundwater water and salt lateral transport, and agricultural water productivity module. Especially, a new method was designed and incorporated to couple regional soil hydrology process and groundwater flow, with the spatial difference of cropping pattern. Taking advantages of remote sensing and GIS tools, the quantitative distributed RIWP model needs fewer soil and groundwater hydraulic parameters and crop growing parameters and only readily available data of several observation sites at the

regional scale, and regional water and salt process can be simulated on a daily time step. Despite the simplifications involved, the proposed methods of irrigation canal and drainage ditches digitization and groundwater-runoff lateral exchange simulation between grids make the spatial IWP simulation in a real distributed way, instead of using a field scale model applied in a distributed mode to simulate all simulation units independently. The calibration and validation results indicates a good performance of RIWP model applied in this typic study area, and spatial distribution of IWP for different crops can be produced.

Programmed in Matlab (Mathworks Inc., 2015), RIWP model can be run on different operating systems. Furthermore, the model includes capability for parallelization of simulations to reduce batch run times when conducting simulations over large areas, conditions, and/or time periods. In the nearly future, enabling the code to be linked quickly with other disciplinary models to support integrated water resource management could be a great improvement of RIWP model. Also, we are going to develop a website used for long-term distribution of the RIWP model and associated documentation. Finally, RIWP model could improve knowledge of best practices to enhance water productivity for key irrigation decision-makers. The simplicity of RIWP model in its required minimum input data, which are readily available or can easily be collected, makes it user-friendly. It is also a very useful model for scenario simulations and for planning purposes, which can be used by economists, water administrators and managers working in the arid irrigated area with shallow groundwater.

## Data availability

The simulation results of the water budget during the simulation period of the JFID in this study are available from the authors upon request (jiyxue@ucdavis.edu).

## Author contributions

JYX and ZLH developed the idea to develop the conceptual RIWP model for irrigated area in arid region with shallow groundwater and complex cropping patterns. JYX wrote the programming code of the RIWP model in Matlab. JYX collected and processed the multiple datasets with the

help of SW, GHH and XX and prepared the paper. The results were extensively commented on

and discussed by ZLH, IW, IK, ZPS, and CZW.

## Competing interests

The authors declare that they have no conflict of interest.

## Acknowledgements

This study was supported by the National Key Research and Development Program of China

(2017YFC0403301), the National Natural Science Foundation of China (51679236, 51639009)

and the International Postdoctoral Exchange Fellowship Program from the Office of China

Postdoctoral Council (20180044). Special thanks also go to the adminstration of Hetao Irrigation

District and Shahaoqu experimental station for providing information and data.

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

Table 1. The significance level of the input parameter to the model output variables

| *SRC* value | Significance level |
|---|---|
| 0.8≤\|SRC\|≤1 | Very important |
| 0.5≤\|SRC\|≤0.8 | Important |
| 0.3≤\|SRC\|≤0.5 | Unimportant |
| 0≤\|SRC\|≤0.3 | Irrelevant |


Table 2. Calibrated crop parameters of wheat, sunflower and maize for regional irrigation water
productivity model

| Parameters | Calibrated value | | |
|---|---|---|---|
| | Wheat | Sunflower | Maize |
| Rate of yield decrease per unit of excess salts, $b$ (%/(ds/m)) | 7.1 | 12 | 12 |
| Average fraction of TAW that can be depleted from the root zone before moisture stress, $p$ (-) | 0.55 | 0.45 | 0.55 |
| Crop coefficient at crop initial stage, $k_{c1}$ (-) | 0.3 | 0.3 | 0.3 |
| Crop coefficient at crop development stage, $k_{c2}$ (-) | 0.73 | 0.8 | 0.75 |
| Crop coefficient at mid-season stage, $k_{c3}$ (-) | 1.15 | 1 | 1.2 |
| Crop coefficient at last season stage, $k_{c4}$ (-) | 0.4 | 0.7 | 0.6 |
| Yield response factor, $K_y$ (-) | 1.15 | 0.95 | 1.25 |
| Electrical conductivity of the saturation extract at the threshold of $EC_e$ when crop yield firstly reduces below $Y_m$ at last season stage, $EC_{et}$ (dS/m) | 5 | 1.7 | 2 |















Table 3. The collected possible parameter variation ranges and calibrated values of the parameters describing soil hydraulic characteristics ($K_e$, $S_y$, $K$) and irrigation and drainage system ($\eta_{lc}$, $\eta_{fc}$, $\gamma_d$, $A$, $m$).

| Parameters | Description | Value range | | Calibrated value |
|---|---|---|---|---|
| | | Min | Max | |
| $K_e$ | Soil evaporation coefficient, (-) | 0.1 | 0.35 | 0.25 |
| $\eta_{lc}$ | Water utilization coefficient of lateral canal, (-) | 0.81 | 0.91 | 0.88 |
| $\eta_{fc}$ | Water utilization coefficient of field canal, (-) | 0.81 | 0.86 | 0.89 |
| $S_y$ | Specific yield, (-) | 0.02 | 0.15 | 0.15 |
| $\gamma_d$ | Drainage coefficient, (-) | 0.02 | 0.06 | 0.03 |
| $K$ | Permeability coefficient of unconfined aquifers, (mm/day) | 731 | 12701 | 1150 |
| $A$ | Soil water permeability coefficient, (-) | 0.7 | 3.4 | 3.4 |
| $m$ | Soil water permeability exponent, (-) | 0.3 | 0.5 | 0.5 |

Note: The parameter value ranges were collected from local measurements, survey data and relevant research results. Soil texture of canal bed was silty sandy loam for 0-1 and 2-3 m depth below the ground, and sandy loam for 1-2 m. For silty sandy loam soil, the bulk density and saturated soil water conductivity are 502.3 mm d$^{-1}$ and 1.42gcm$^{-3}$, respectively. For sandy loam soil, the bulk density and saturated soil water conductivity are 1.49g cm$^{-3}$ and 592.6 mm d$^{-1}$, respectively. There were fine sand and sandy soil in the phreatic layer.

**Figure Captions**
**Fig.1.** Schematic diagram of the conceptual RIWP model and the coupling between its sub-
modules.
**Fig.2.** Schematic diagram of groundwater lateral runoff exchange between HRUs.
**Fig.3.** Schematic diagram of coupling soil water and salt dynamics, and groundwater level and
salinity. And the IWP evaluation in each HRU.
**Fig.4.** Procedure chart of regional irrigation water productivity simulation.
**Fig.5.** Location of the Jiefangzha Irrigation District.
**Fig.6.** Relationship between the simulated and measured values during the crop growing season in
calibration and validation period.
**Fig.7.** The comparison of the simulated and measured groundwater table depth for 4 typical sites
during the crop growing season in the years of 2006-2013. (Note: a- uncultivated area during the
years of 2006-2013; b- uncultivated area from 2006-2008, and sunflower field and maize field
from 2009-2013; c, d- sunflower, wheat and maize field in the years of 2006-2013)
**Fig.8.** Parameter sensitivity analysis results of model for the three output variables: (a) irrigation
water productivity, (b) groundwater table depth and (c) groundwater salinity.
**Fig.9.** Spatial distribution of irrigation water productivity for the three main crops during the
period of 2006-2013. Each line shows the RIWP for each year by ascending order. The left, middle
and right column shows the RIWP of wheat, sunflower and maize, respectively.
**Fig.10.** (a) Simulated regional irrigation water productivity under various groundwater table depth
($h_g$) conditions with different irrigation water amount ($I_n$) applied, and (b) its statistical analysis
results. In Fig.10a, W, S and M represents wheat, sunflower and maize, respectively






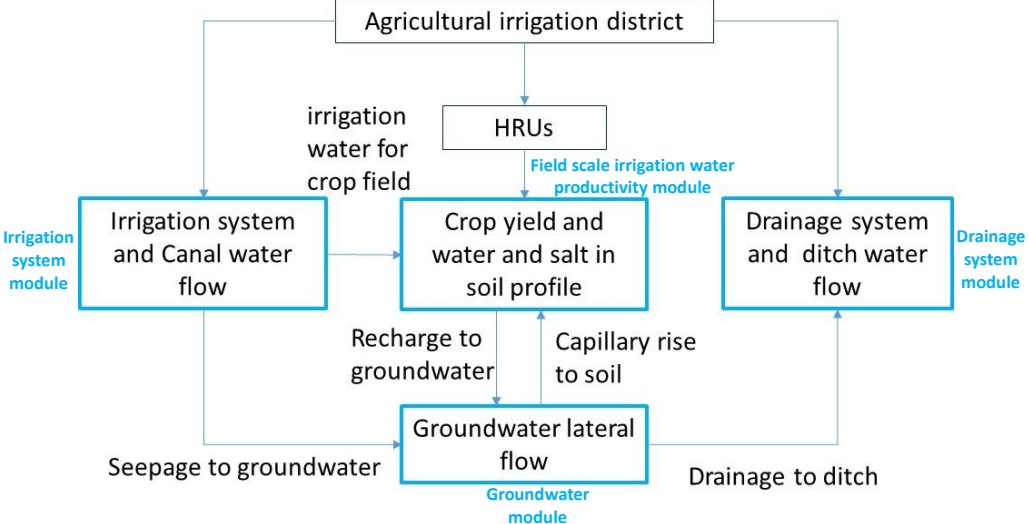


**Fig.1.** Schematic diagram of the conceptual RIWP model and the coupling between its sub-
modules.

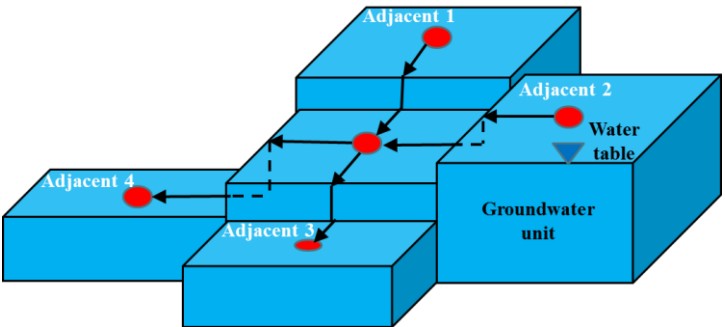


**Fig.2.** Schematic diagram of groundwater lateral exchange between adjacent HRUs.

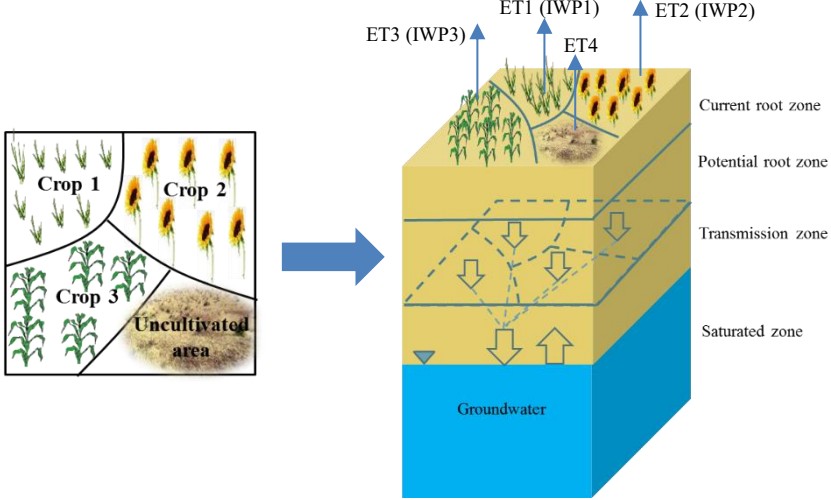


**Fig.3.** Schematic diagram of coupling soil water and salt dynamics, and groundwater level and
salinity. And the IWP evaluation in each HRU.

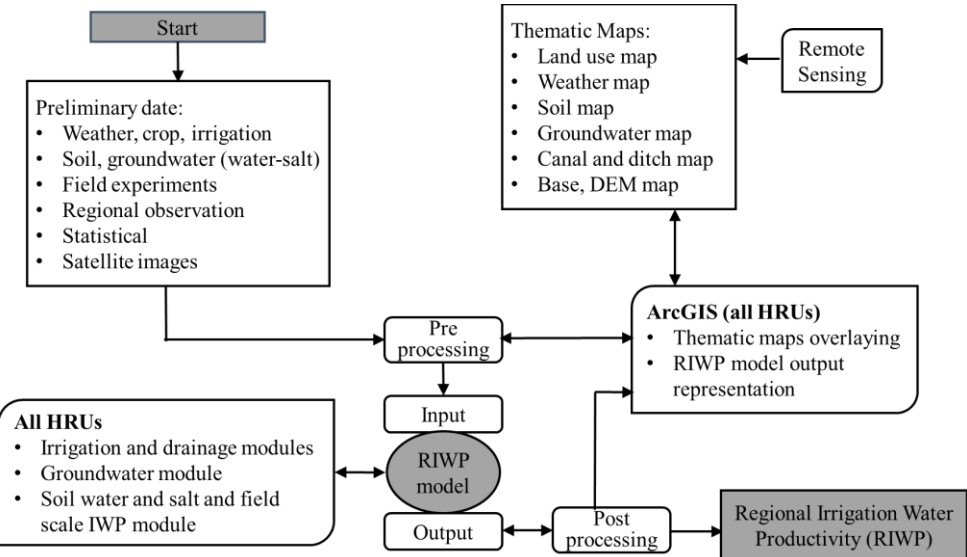


**Fig.4.** Procedure chart of regional irrigation water productivity simulation.


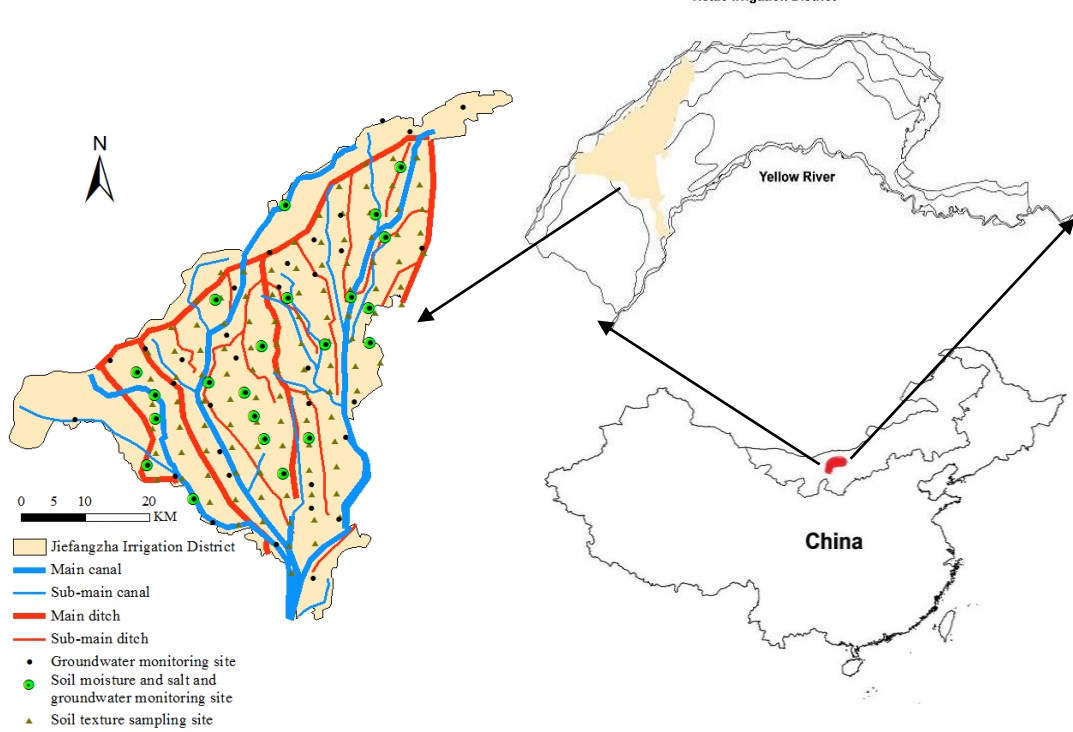


**Fig.5.** Location of the Jiefangzha Irrigation District.





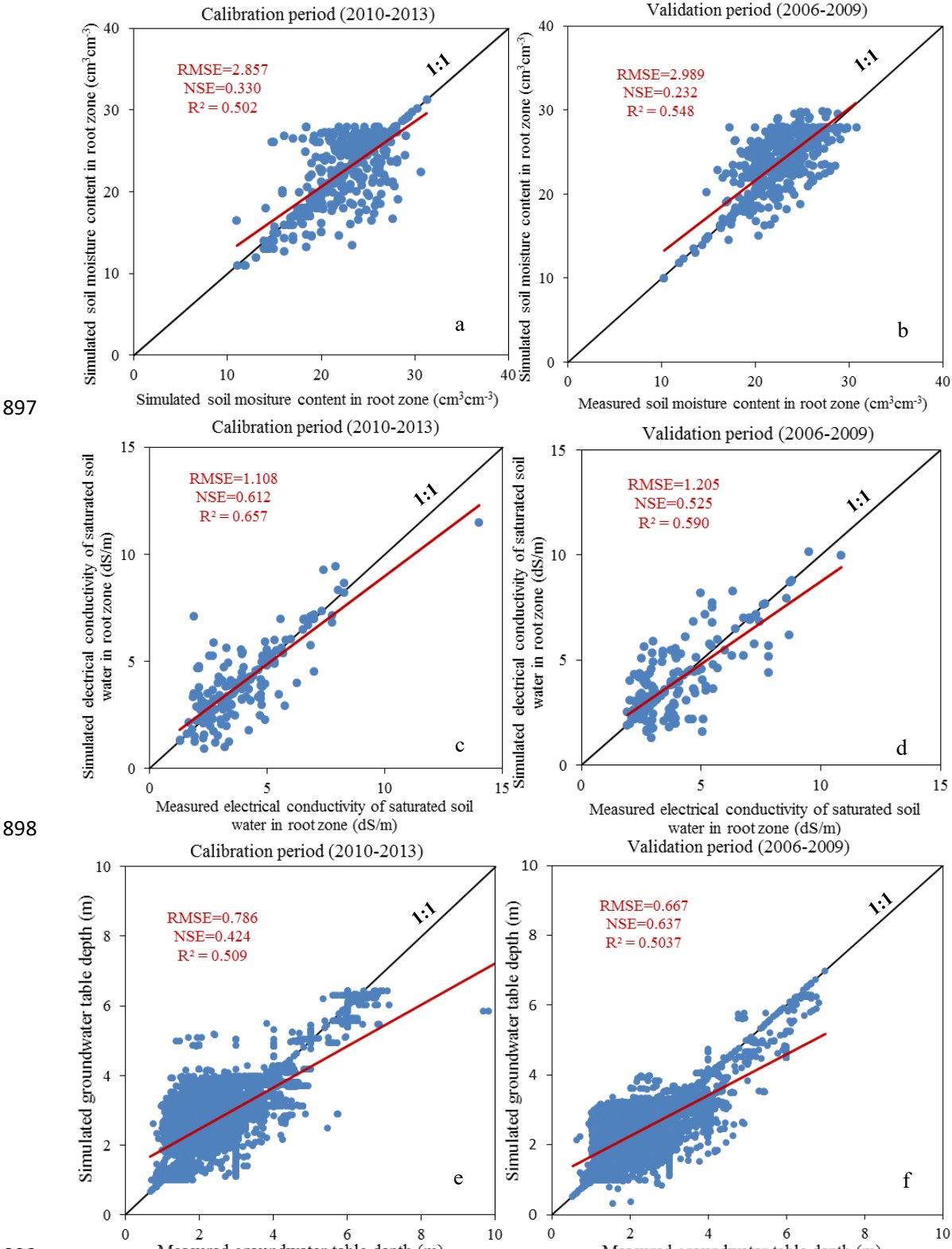



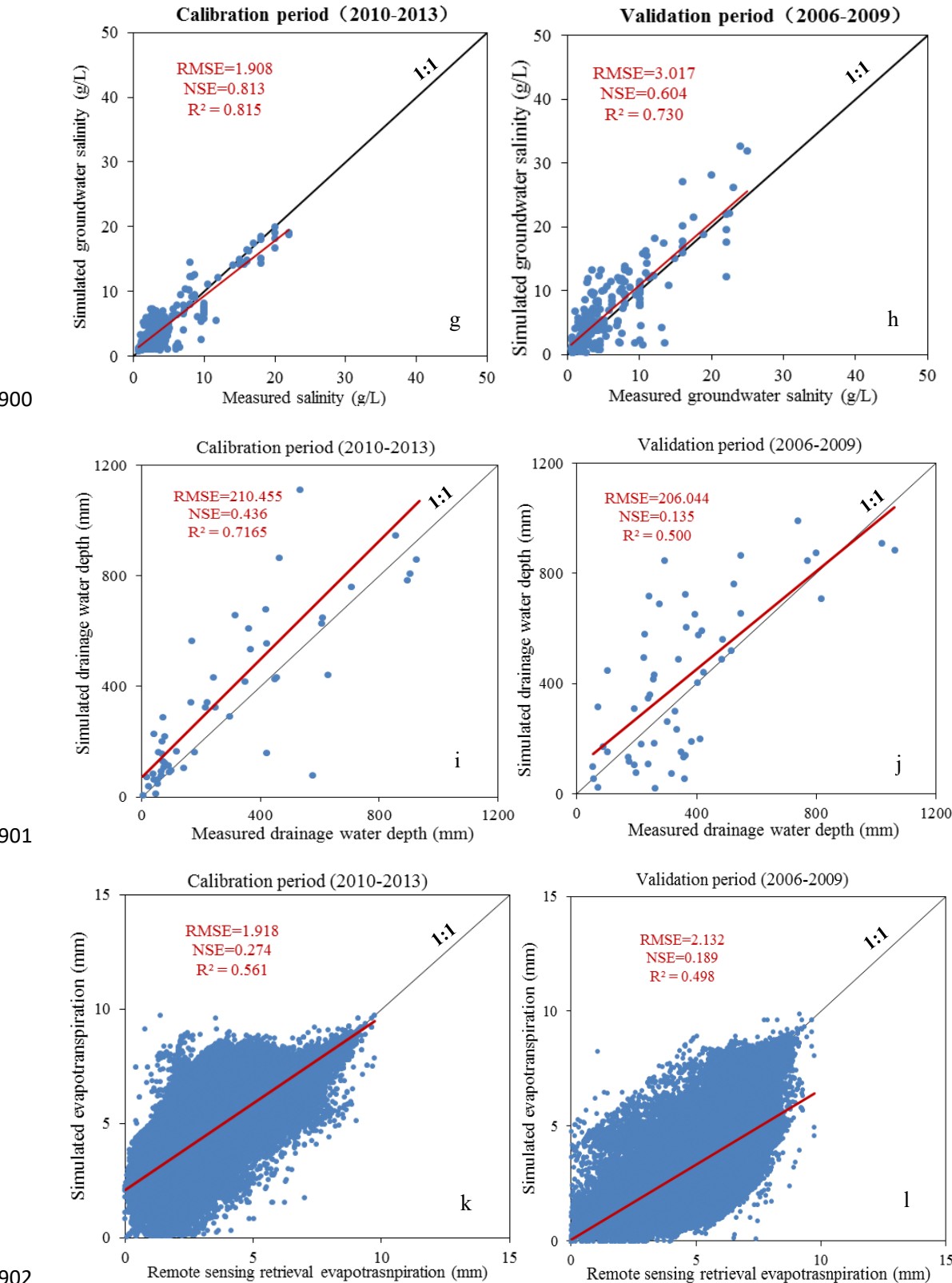




**Fig.6.** Relationship between the simulated and measured values during the crop growing season in
calibration and validation period.

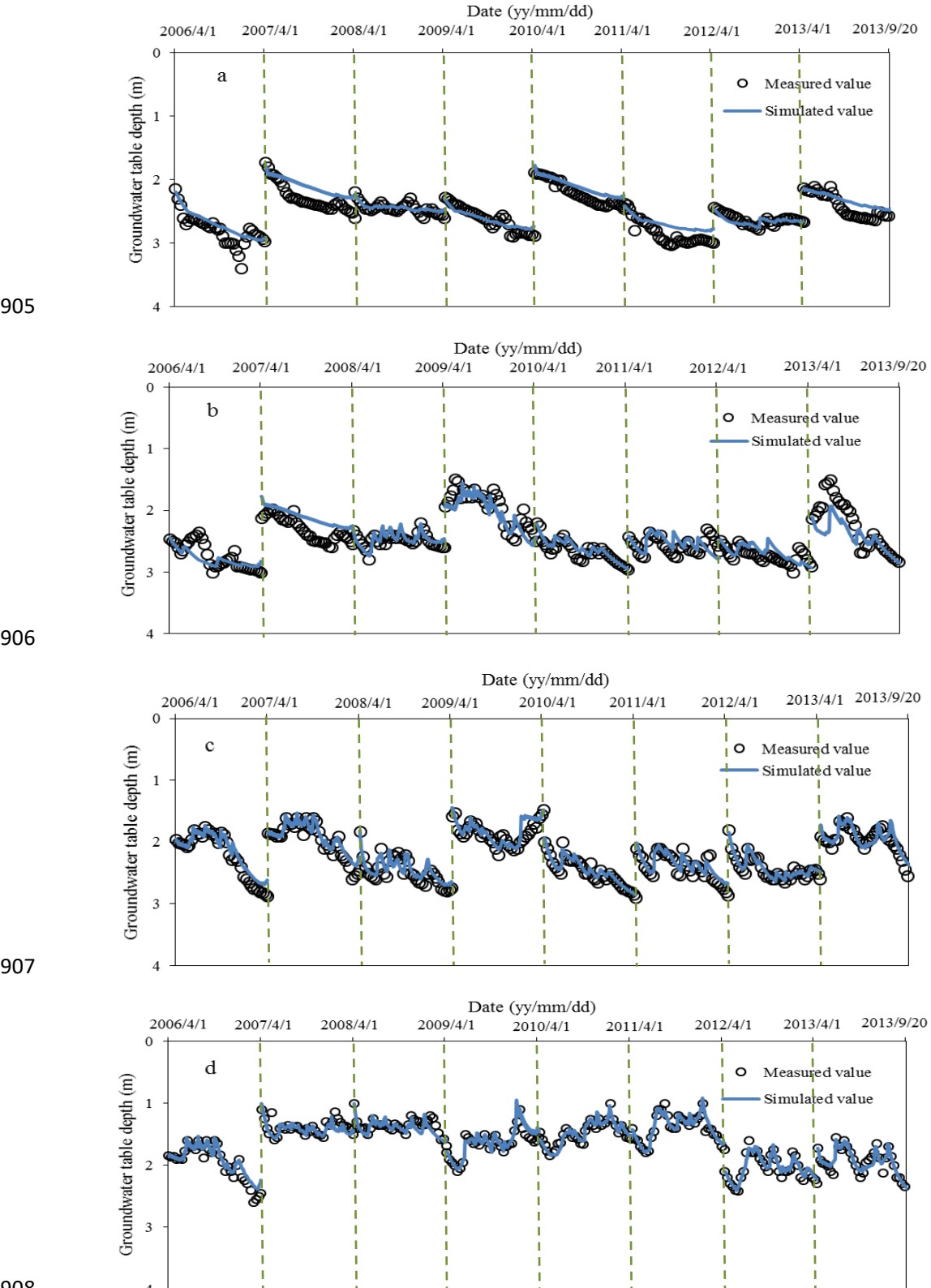





**Fig.7.** The comparison of the simulated and measured groundwater table depth for 4 typical sites

during the crop growing season in the years of 2006-2013. (Note: a- uncultivated area during the

years of 2006-2013; b- uncultivated area from 2006-2008, and sunflower field and maize field

from 2009-2013; c, d- sunflower, wheat and maize field in the years of 2006-2013)


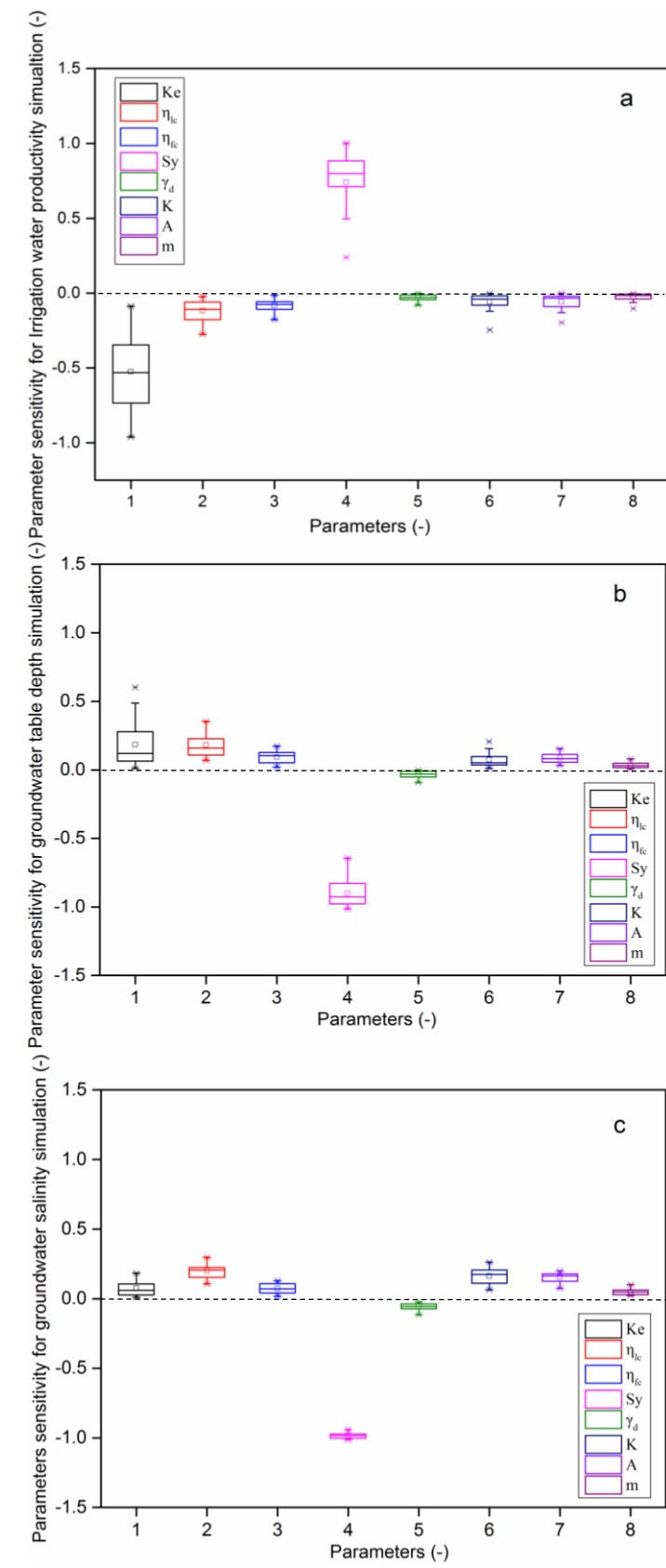


**Fig.8.** Parameter sensitivity analysis results of model for the three output variables: (a) irrigation
water productivity, (b) groundwater table depth and (c) groundwater salinity.

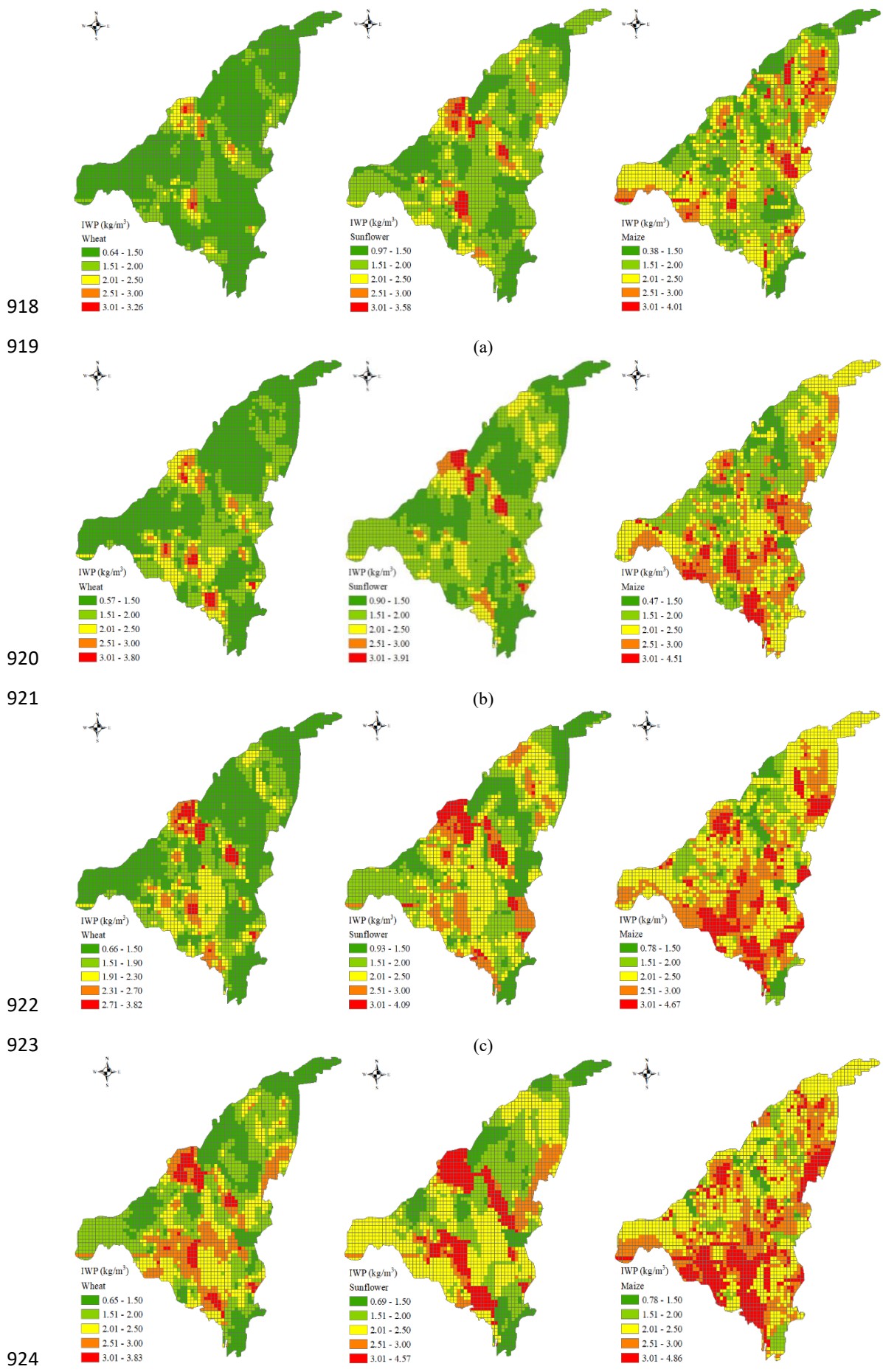


(a)


(b)


(c)



(d)

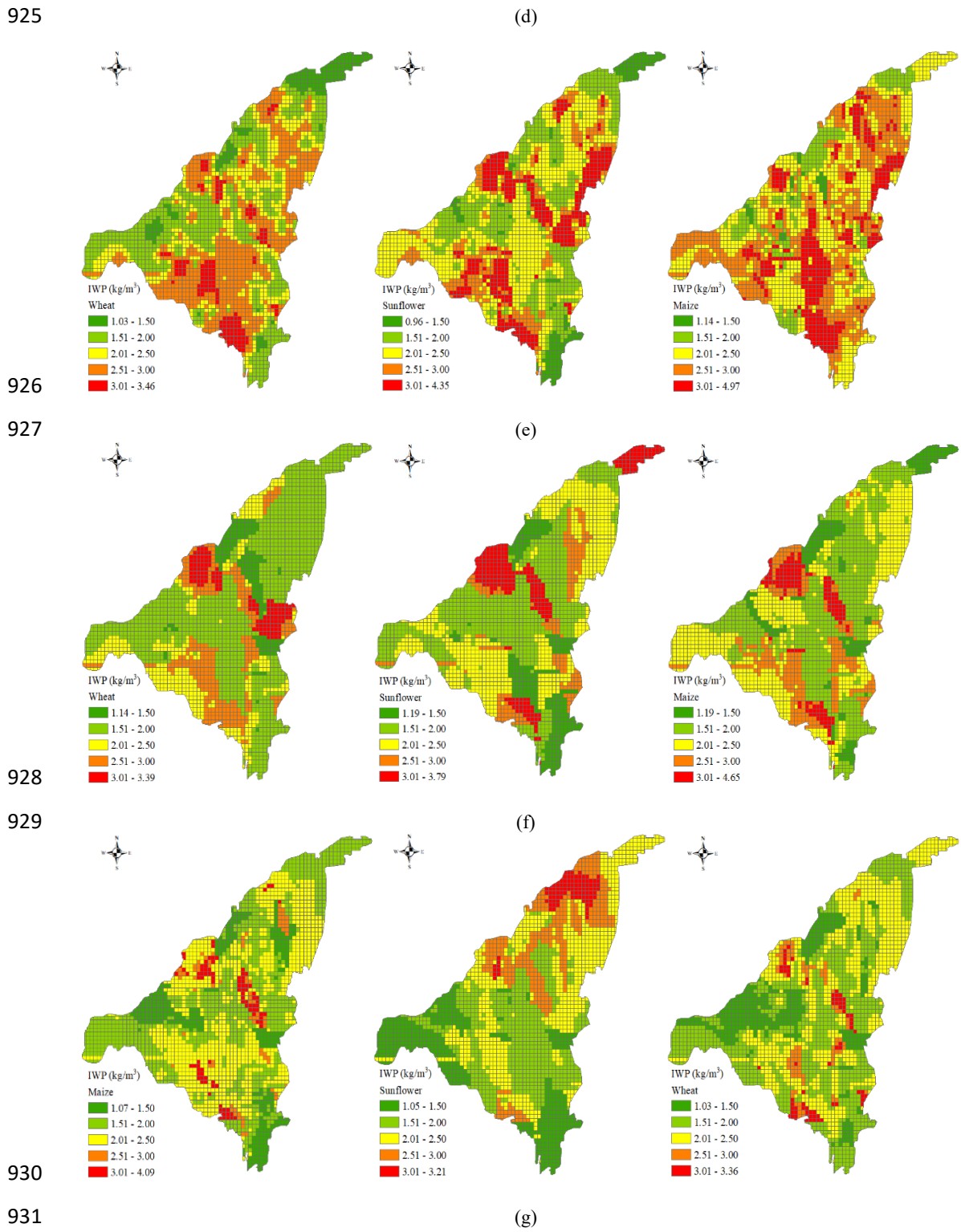



(e)



(f)



(g)

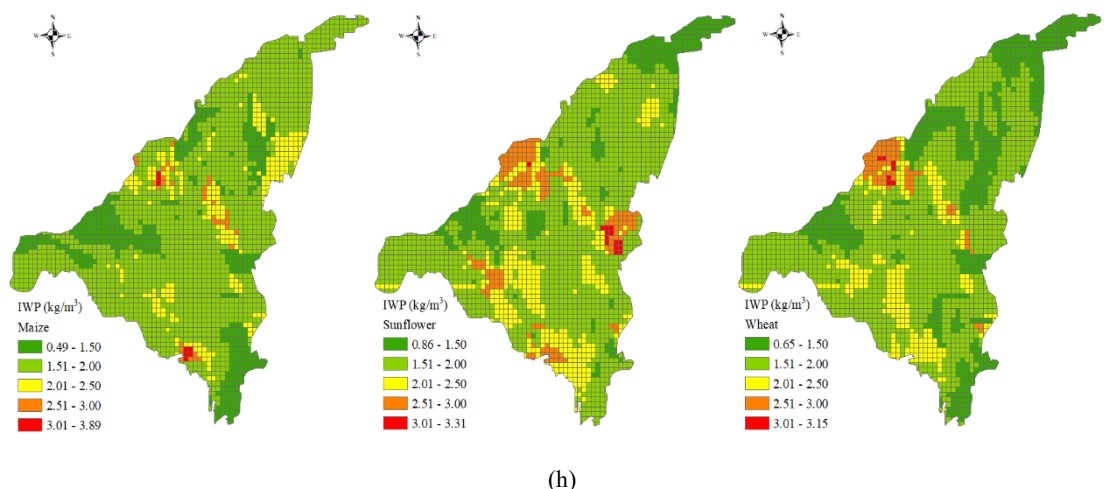

(h)

**Fig.9.** Spatial distribution of irrigation water productivity for the three main crops during the period of 2006-2013. Each line shows the RIWP for each year by ascending order. The left, middle and right column shows the RIWP of wheat, sunflower and maize, respectively.

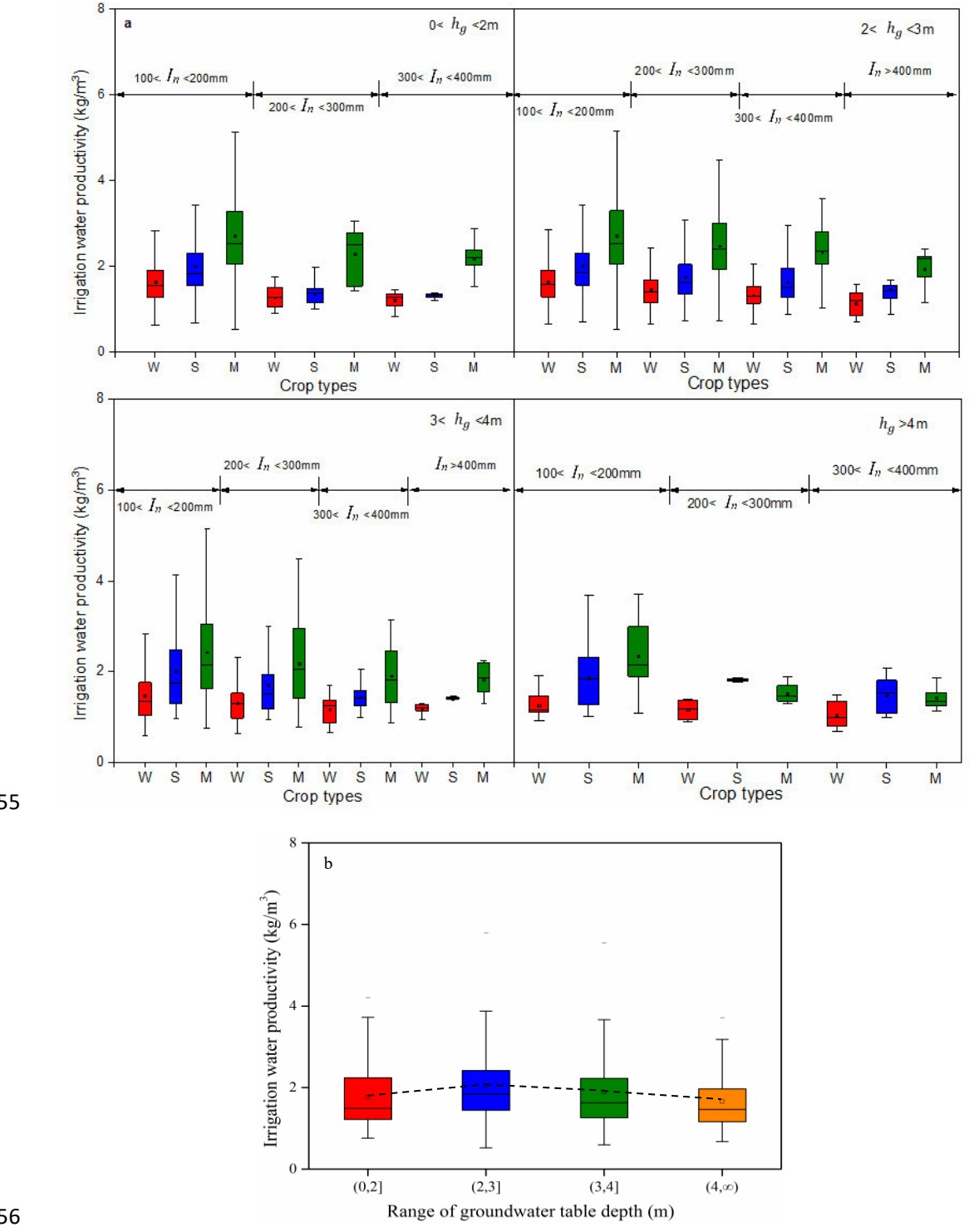

955

956

**Fig.10.** (a) Simulated regional irrigation water productivity under various groundwater table depth
($h_g$) conditions with different irrigation water amount ($I_n$) applied, and (b) its statistical analysis
results. In Fig.10a, W, S and M represents wheat, sunflower and maize, respectively.