# Peer review of "A novel regional irrigation water productivity model coupling irrigation-drainage driven soil hydrology and salinity dynamics, and shallow groundwater movement in arid regions, China"

_Hydrology and Earth System Sciences, 2019_

## Referee Comment (RC1) · Zhongyi Qu (Referee) · 13 Nov 2019

Comments:

The study principally simulated soil hydrology and crop irrigation water productivity with recently developed regional temporal-spatial hydrological model in the arid district. These results attributes mainly to the dynamic-management of local agricultural water resources distribution and crop cropping system under changing climate environment,

e.g. salinity, groundwater depth. The paper is well written and organized with novel idea and new findings. The model's simulation results are reasonable. Suggest accept after addressing these comments:

1. The title is too long and needs revision. Suggest: A novel regional irrigation water productivity model coupling soil hydrology and salinity dynamics in arid regions, China 2. L39-40 in Abstract, how about the simulation agreement of validation and calibration plots? 3. Provide details on model's calibration procedure before L345 as subtitle 2.3.2. 5. Crop growth is closely with ET? What are the model simulation performances of cash crops growth (biomass, LAI, phonology) and grain yield in the calibration and validation systems in the section of 3.1. 4. Each section of the three Results and Discussion is needed for greater improvement especially in global sensitivity analysis and irrigation water productivity. Provide more explanations regarding the cause of simulation results, except for comparison with similar previous study results. 5. L705, what are the measured values? Detail on figure title. 6. L704 provide details on soil particle size, bulk density, saturated water conductivity in table 3 7. Figure 10, there was no obvious difference in irrigation water productivity in groundwater 0-1 and 1-2 m? If not, provide the corresponding results between these groundwater levels

Please also note the supplement to this comment:
https://www.hydrol-earth-syst-sci-discuss.net/hess-2019-359/hess-2019-359-RC1-supplement.pdf

---

## Referee Comment (RC2) · Anonymous Referee #2 · 14 Feb 2020

Review of "A novel regional irrigation water productivity model for complex cropping patterns in arid regions coupling soil water and salinity dynamics, irrigation and drainage, and shallow groundwater movement"

Recommendation I like this paper and believe it should be published with medium and minor edits. It is well-written and structured but will need some copy-editing as some of the English grammar and syntax can be improved. The main changes should relate

to how the authors can make their model and its results more reader-friendly in that readers will want to know how this model helps users and managers better manage irrigation water. With the current version, it is not clear at the moment where these insights sit. In other words the author's own interpretation of their RIWP model needs to be more clearly written.

Substantive comments The productivity model depends on four parameters; water supply from irrigation open canals, field crop water consumption, groundwater drainage into open ditches, and groundwater lateral flow. Can the authors explain why rainfall is not included in their model as a water supply to crop growth? How would the model work in an area with more rainfall than in their case study?

Can the authors explain why lateral movement between drainage 'bonds' the units together (line 160) but that lateral movement of irrigation water down channels does not? Surely irrigation water and drainage water are both moving laterally as well as vertically?

Seepage loss from channels is in the model, but I do not readily spot where seepage loss beneath the root zone from fields is accommodated?

The authors write on page 17 a statement that the contribution of groundwater and proportion of non-beneficial soil evaporation are major influences on water productivity of their chosen crops. This seems to indicate that the productivity model is simply a biomass model related to the proportion of total water supply that ends up in transpiration? But there are other factors such as irrigation timing and scheduling that affect productivity. This makes this reviewer wonder what are the units of RIWP? And why are these units not utilised frequently throughout the paper? Thus in other words is this a production model not a productivity model?

Also can the authors explain why, if nearly all the groundwater supplies and movements of water which are in the model come from irrigation both in the short and long-term, and not from rainfall or wider hydrogeological inflows, does the model 'bother' with

groundwater as a factor determining IWP? Surely the main determinant of irrigation productivity in an entirely arid region is really only 'irrigation supply'. This reviewer knows partly the answer but the authors must not assume the readers know this distinction.

Line 490 – can the authors explain why productivity declines when water supply from irrigation goes up? This may be consistent with other results, but it is counter to expectation? (Again the problem is that the units of IWP are not given in the main body of the paper).

Can the authors be clear about what m3 of water on the denominator is about – is it total supply in cubic metres or is it total transpired cubic metres?

As a key comment, I think Section 3 needs to be re-written by starting or leading with key management results and insights that are readable by different stakeholders. At the moment this section is written with the model rather than the results in mind. The key management insights are buried deep within this section and are not easy to find. Here are some guide questions that show what I mean:

Which affects crop productivity more – irrigation dose/depth applied or the contribution from groundwater?

Which affects crop productivity more – lots of shallow irrigation applications or fewer deeper applications?

Which type of crop is most productive in coping with water supply coming from non-irrigation sources?

How is productivity negatively or positively affected by a combination of drainage and salts?

What explains the changing 'red spots' of high productivity in the maps in Figure 9 and whether and how this high level of productivity can be extended to the rest of the Jiefangzha Irrigation District so that everything becomes 'red'.

I hope these examples show why the 'results' section currently does not clearly guide managers and planners.

Can the authors also introduce some 'future or methodological critical thinking'? In other words, how does such an approach really guide current managers in improving irrigation management? What future improvements to the method and model might allow this to happen? How does the author's model differ from other regional irrigation productivity studies, eg. conducted by the Water for Food Institute, Nebraska.

Minor comments

Be consistent "water productivity model" in title, but "water productivity estimation" in key words.

Is there a substantive difference between "irrigation water productivity (IWP)" and "regional irrigation water productivity (RIWP)"

Line 36. Are uncultivated lands bare lands, or natural vegetation?

Line 45. I would use the words 'depth applied' or 'delta and deltas' when discussing water applied via irrigation (and not 'depth' alone). Otherwise this use is confusing "when groundwater table depth is in the range of 2 m to 4 m, regardless of irrigation water depths"

Line 54. I would not use a single figure of 90% here "where irrigated agriculture accounts for about 90% of the total". I would use a range e.g. 70 to 90%

Line 69 Field experiments may be costly but they do allow for calibration and an understanding of the relevant parameters and processes "but field experiments are expensive and time consuming, making it unsuitable for regional evaluation of IWP." So field experiments still help with a regional evaluation?

Line 84, can an example of simplified distributed models be given? "There are two types of distributed hydrologic models that are used to integrate with crop models: numerical distributed models, such as SWAT and MODFLOW, and simplified distributed models based on water balance equations."

Line 91. Where in the paper is this significance of groundwater explained – not all arid and semi-arid areas have shallow groundwater? Do the authors take this for granted? " SWAT alone does not describe the complex interactions between groundwater and soil water, which are fundamental in arid and semi-arid areas with shallow groundwater".

Line 94 – suggest small change "However, the large spatial grids can hardly reflect the regional complex cropping pattern heterogeneity, and the large temporal steps cannot capture daily soil water" to this "However, the large spatial grids poorly reflect the re-gional complex cropping pattern heterogeneity, and the large temporal steps cannot capture daily soil water"

Line 139 The authors could do better in explaining what an HRU is? Is it an abstract artefact, or a real command unit within an irrigated landscape? Do irrigation managers use HRUs?

Line 230 can this sentence about boundaries be explained? "There are three types of groundwater boundaries: river boundaries, drainage ditch boundaries and no flux boundaries"

Line 258 spelling/grammar? "Cropping patterns are complex for each HRU and some-times HRU include uncultivated land, forest". This should be "Cropping patterns are complex for each HRU and sometimes HRUs include uncultivated land, forest"

Line 293 – correct this sentence to "Considering the high spatial heterogeneity, mete-orological data need to be collected from all the weather stations within or close to the study area."

Line 427 check grammar to this "the ditches of the same order share the same the drainage coefficient, assuming well-operated conditions. However,"

Line 502 – difficult to follow the argument with the current English. Should this

not read "indicates that when irrigation applied decreased from 300<IWD<400mm to 200<IWD<300mm it lead to decreases in IWP caused by a reduction of ET." (But this seems to contradict statements made elsewhere in the paper?

Line 505 onwards – very difficult to understand this text! "ET, which is less irrigation water will weaken the role of irrigation on salt leaching and result in more severe salinization in crop root zone. Thus, reasonably determining the irrigation quota and constantly maintaining the drainage system to keep the groundwater table depth in the optimal range is of great importance to reach higher crop IWP at the regional scale."

Line 511. Does not make sense "In view of the particularity of irrigated areas, taking fully consideration of the supply," Perhaps this? "In view of the heterogeneous conditions of irrigated areas, taking fully consideration of the supply,"
* * *

---

## Author Response (AR1)

**Revision Notes (hess-2019-359)**

**Responses to the comments of Editor:**

**Recommendation:** The manuscript has been reviewed by two reviewers, and the authors have, in my view, responded adequately to the issues raised.

I have, however, two additional issues that were not raised by the reviewers, and which I like to share with the authors. I invite the authors to take my comments into account when submitting an improved version of the paper. I encourage the authors to submit a revised version of the paper, taking also the above details into account.

**Response:** We are appreciating to the editor for the useful comments and suggestions to the paper. Based on that, we have made corresponding changes to furtherly improve the quality of this paper. Below are the detailed responses to all comments. We cited first the comment, which is followed by our response and often by a section how the text will be revised in the manuscript. The text in blue are changes and additions in the original text. For clarity we do not show the removed text in the blue content.

**Comment1:** There are some references in the text that do not appear in the reference list; take for example the references cited in lines 55 to 62: of the 9 references, 5 do not appear in the reference list. Also check the reference in line 174.

**Response:** Thanks very much for this useful comment. We are sorry for not presenting the references in the reference list. Here we added corresponding references in L672-673, L694-696, L709-713, L742-744, L765-766 as following:

"Bouman, B. A. M., 2007. Water management in irrigated rice: coping with water scarcity. Int. Rice Res. Inst.."

"Jiang, Y., Xu, X., Huang, Q., Huo, Z., Huang, G., 2015. Assessment of irrigation performance and water productivity in irrigated areas of the middle Heihe River basin using a distributed agro-hydrological model. Agricultural water management, 147, pp.67-81."

"Men, B. H., 2000. Discussion on formula of channel flow loss and water utilization coefficient.

China Rural Water and Hydropower, 2, 33-34.

Morison, J.I.L., Baker, N.R., Mullineaux, P.M., Davies, W.J., 2008. Improving water use in crop production. Philosophical Transactions of the Royal Society B: Biological Sciences,

363(1491), pp.639-658."

"Surendran, U., Jayakumar, M., Marimuthu, S., 2016. Low cost drip irrigation: Impact on sugarcane yield, water and energy saving in semiarid tropical agro ecosystem in India.

Science of the Total Environment, 573, pp.1430-1440."

"Williams, W.D., 1999. Salinisation: A major threat to water resources in the arid and semi-arid regions of the world. Lakes & Reservoirs: Research & Management, 4(3-4), pp.85-91."

**Comment2:** The manuscript is inconsistent with its units. All water fluxes should have a time dimension. So $W_{ls}$ (line 193-194) is the groundwater recharge per unit; and in your model you use a daily time step, so the correct unit is m/day. Same for $W_{as}$ (line 202-203), $I_n$ (line 203), $D_g$ (line

213), $W_{gr}$ (line 224), $P_{wg}$ (lines 251-252), $G_{wg}$ (lines 252-253). Check the correct unit of $K$

(permeability coefficient, lines 224-225), I think it should have a time dimension. Check eq.10 on consistency of the units/dimensions.

**Response:** Thanks very much for this useful comment. We are sorry for careless writing on the units of all water fluxes. Here we did a throughout check on all water fluxes' units and made corresponding corrections in L203-204, L212-214, L224, L227-228, L235-236, L255-258, L266-

269 of revised manuscript as following:

"$W_{ls}$ is the daily groundwater recharge per unit area due to water conveyance loss in main and sub- main canals (mday$^{-1}$)."

"where $W_{as}$ represents daily groundwater recharge per unit area due to water conveyance loss in lateral and field canals (mday$^{-1}$), and $I_n$ is daily irrigation water depth applied per unit area (mday$^{-}$

$^{1}$)."

"where $D_g$ is daily groundwater drainage per unit area (mday$^{-1}$)."

"$h_g$ represents the daily groundwater table depth (mday$^{-1}$), and $h_{db}$ is the daily streambed depth of drainage ditch (mday$^{-1}$)."

"where $W_{gr}$ is the daily groundwater inflow of the current HRU from adjacent HRUs (mday$^{-1}$), and

$K$ is the daily permeability coefficient of unconfined aquifers in the current HRU (mday$^{-1}$)."

"where $W_{grup}$, $W_{grdown}$, $W_{grleft}$ and $W_{grright}$ are the daily groundwater lateral runoff per unit area into the current groundwater unit from up and down or left and right adjacent groundwater unit, respectively (mday$^{-1}$). $SCa$ is the daily soluble salt content in the saturated zone below the transmission soil profile (mg m$^{-2}$day$^{-1}$)."

"$ext$ is the daily groundwater extraction per unit area (mday$^{-1}$). $P_{wg}$ is the daily percolation water depth to groundwater from the potential root zone (mday$^{-1}$), and $G_{wg}$ is the daily water depth supplied to the potential root zone from shallow groundwater due to the rising capillary action (mday$^{-1}$). $P_{sg}$ and $G_{sg}$ are the quantity of soluble salt in $P_{wg}$ and $G_{wg}$, respectively (mg m$^{-2}$day$^{-1}$)."

**Comment3:** The amount of irrigation water applied seems small (lines 320-323); I calculated an average gross irrigation application of 162 mm/year [(12x108)/(0.66*1.12*106*104)=0.162

m/year]. Kindly explain.

**Response:** Thanks very much for this useful comment and suggestion. We are sorry for the careless writing about the area of JFID, and the correct number should be 0.22 Mha. We made corresponding correction in L336 of revised manuscript as following:

"The JFID covers an area of 0.22 Mha…"

**Comment4:** Lines 388-391: What are thresholds for acceptable and good model performance for the 3 evaluation criteria used (NSE, R2 and RMSE)?

**Response:** Thanks for this useful comment. We made further explanation of the thresholds for acceptance and good model performance for these three evaluation indexes in L410-419 of revised manuscript as following:

"The *RMSE* indicates a perfect match between observation and simulation when it equals 0, and increasing *RMSE* values indicate an increasingly poor match. Singh et al. (2005) stated that *RMSE*

values less than 50% of the standard deviation of the observed data could be considered low enough as an indicator of a good model prediction. Ranging between $-\infty$ and 1, the NSE

indicates a perfect match between observed and predicted values when it equals to 1. Values between 0 and 1 are generally considered as acceptable levels of performance, whereas values less than 0.0 indicate that the simulation is worse than taking an average of observation, which indicates unacceptable performance. The $R^2$ ranging between 0 and 1 describes the proportion of the variance in the observed data, in which higher values indicating less error variance. Typically,

$R^2 > 0.5$ is considered acceptable (Santhi et al., 2001)."

Additionally, we added two references in the reference list in L724-726 and L733-735 of revised manuscript as following:

"Santhi, C., Arnold, J.G., Williams, J.R., Dugas, W.A., Srinivasan, R., Hauck, L.M., 2001.

Validation of the swat model on a large rwer basin with point and nonpoint sources 1.

JAWRA Journal of the American Water Resources Association, 37(5), pp.1169-1188."

"Singh, J., Knapp, H.V., Arnold, J.G., Demissie, M., 2005. Hydrological modeling of the Iroquois river watershed using HSPF and SWAT 1. JAWRA Journal of the American Water

Resources Association, 41(2), pp.343-360."

**Comment5:** Line 451: "readily available groundwater"; here I think you deal with the unsaturated zone, so do you rather mean: "readily available soil moisture"?

**Response:** Thanks very much for this comment and suggestion. We actually tried to express the parameter specific yield as the volume of water gained or lost under gravity or capillary action with a corresponding amount of water table rise or fall. Here we corrected the expression of sentence in L490-492 of revised manuscript as following:

"The specific yield indicated the readily available soil moisture released to crop root zone from shallow aquifer under capillary action for crop consumption."

**Comment6:** Figure 9: in an earlier iteration I asked the authors to improve the colour-scheme of this figure. You have done so, but in the process, you have, unfortunately, not standardized the scales (as you had done in the original version of this figure, and as you have also correctly done in your figure S3). For the reader it is therefore very difficult to compare the different years. So for each crop redraw the maps by keeping the colour scale fixed over the years.

**Response:** Thanks very much for this comment and suggestion. We redrew the IWP maps for three main crops in Fig.9 to make the scales standardized. After that, we believe that readers could compare the different years of IWP for three crops easily. Detailed changes see Fig.9 in the revised manuscript.

**Comment7:** Figure 9 once more: none of the maps contain blank pixels – this suggest that each pixel in all years have values for the productivity of all three crops. This I find highly surprising, and in fact unlikely, (but I admit that I do not know the irrigation district). Please explain.

**Response:** Thanks for this useful comment. Fig.9 represents the spatial distribution of IWP for three crops (wheat, maize and sunflower) at a given year at 1km*1km simulation unit scale. As you know, although main crops is wheat, maize and sunflower, spatial distribution of these crops is very complex and field plot is small.   we use remote sensing data to get cropping pattern map with resolution of 30m*30m, almost every HRU (1km*1km) have these crops. Thus, we can simulate IWP for each main crop in every HRU. Considering the heterogeneity of cropping pattern in the simulation unit, therefore, even if there is only one pixel of any crop planted in the

1km*1km simulation unit, the IWP of this crop should be reflected on the current simulation unit in the RIWP map in Fig.9. Fig. 9 only shows the IWP of crop located in related HRUs. Thus, we would like to keep our original Figure 9 in the revised manuscript expect for standardizing the scales. We have explained these in L526-529 of revised paper as following:

"As we mentioned before, the spatial distribution of these three crops is very complex in JFID and field plot is small, thus we use remote sensing data to obtain cropping pattern map with resolution of 30m*30m. Every HRU has these three crops, thus we can simulate IWP for each main crop in every HRU."

**Comment8:** Section 3.2.1 concludes about which crops have the highest productivity (lines 481-

486). Here productivity in money value (expressed e.g. in US$/m3 or RMB/m3) would be the most convincing criterion. Do you have average farm gate prices of the three crops, so that you can convert the IWP (kg/m3) into RMB/m3? You suggest that sunflower has a much higher

"benefit" (line 485) than wheat. Do you mean "price"?

**Response:** Thanks very much for these useful comments. Yes, "higher benefit" here indicated that sunflower has a much higher "price" per unit weight than the other two crops. We are currently working on another paper which is focused on addressing how productivity in money value varied under the effect of years of water saving agricultural development in JFID. Here, in this paper, we would like to focus on looking at the simulation result of our RIWP model, which is the IWP, crop yield per cubic meter of irrigation water applied.

**Comment9:** The manuscript still is weak in grammar, and reviewer #2 did a great job to highlight the major weaknesses. Please also check the following lines: 15, 72, 98, 146, 196, 206, 269, 293,

295, 296, 315, 364, 427.

**Response:** Thanks very much for this comment. We made further correction on writing to improve the quality of this paper. Below are detailed revised places:

L15-16 "Department of Land, Air and Water Resources & Department of Biological and

Agricultural Engineering …"

L77: "However, remote sensing is looking at seeing…"

L106: "…productivity models in irrigated areas"

L157: "…first runs field IWP model"

L206-207: "Lateral and field canals are densely distributed in the irrigated area, and they are intermittently filled with low water flow."

L217: "In the drainage system module, only the groundwater draining into ditches is considered. …"

L285: "Finally, the weighted averages are used to update daily groundwater …"

L310-312: "Distribution of soil physical properties, moisture and salinity in unsaturated soil, groundwater table depth and salinity, need to be collected from many observation sites, which are uniformly or randomly spread over the study area."

L330-331: "…arid irrigated area with shallow groundwater, resulted from its arid-continental climate, over years of flood irrigation, and poor drainage systems"

L383-384: "…, which covers the growing seasons of all the three main crops."

L459-460: "In the model, for each year, we adopt same drainage coefficient for all the ditches of the different orders, assuming a well operated condition."

**List of all relevant changes corresponding to the comments of Editor:**

[revised manuscript text omitted]

**Responses to the comments of Reviewer #1:**

The study principally simulated soil hydrology and crop irrigation water productivity with recently developed regional temporal-spatial hydrological model in the arid district. These results attributes mainly to the dynamic-management of local agricultural water resources distribution and crop cropping system under changing climate environment, e.g. salinity, groundwater depth. The paper is well written and organized with novel idea and new findings. The model's simulation results are reasonable. Suggest accept after addressing these comments:

**Response:** We are appreciating to the reviewer for the useful comments and suggestions to the paper. According to your comments, we have made further efforts to make the paper acceptable for publication. We make a large number of revisions based on the comments to make the paper easier to read. We believe that the quality of this paper has been fundamentally improved after that.

Below are the corresponding responses to the reviewer's eight detailed comments. We cited first the comment, which is followed by our response and often by a section how the text will be revised in the manuscript. The text in blue are changes and additions in the original text. For clarity we do not show the removed text in the blue content.

**Comment1:** The title is too long and needs revision. Suggest: A novel regional irrigation water productivity model coupling soil hydrology and salinity dynamics in arid regions, China

**Response:** Thanks very much for this useful comment. We rewrote the title to "A novel regional irrigation water productivity model coupling irrigation-drainage driven soil hydrology and salinity dynamics, and shallow groundwater movement in arid regions, China".

**Comment2:** L39-40 in Abstract, how about the simulation agreement of validation and calibration plots?

**Response:** Thanks very much for this useful comment. We added the detailed model simulation performance in the L41-45 of revised manuscript as "The model reasonably well simulated soil moisture and salinity, as well as groundwater table depths and salinity. Overestimations of groundwater discharge were detected in calibration and validation due to the assumption of well-operated condition of drainage ditches, and regional evapotranspiration (ET) were reasonably estimated while ET in uncultivated area was slightly underestimated in RIWP model".

**Comment3:** Provide details on model's calibration procedure before L345 as subtitle 2.3.2.

**Response:** Thanks very much for this useful comment and suggestion. We added the detailed procedures of model's calibration and validation procedures in the revised manuscript as subtitle 2.3.3 as following:

**2.3.3 Model calibration and validation**

To comprehensively evaluate the accuracy and reliability of the model, the data in years 2010-2013 and in years 2006-2009 was respectively used as calibration and validation dataset. The daily measured soil moisture content of crop root zone ($\theta$), electrical conductivity of soil water (EC), groundwater table depth ($h_g$) and groundwater salinity, were calibrated with measured data from the 22 soil water and salt observation sites and 55 groundwater observation sites (Fig. 5), which were mentioned in section 2.3.1. The RIWP simulated regional ET for each HRU was calibrated by the remote sensing based ET images obtained once per 8 days. The regional drainage processes was calibrated by the monthly groundwater drainage data from main ditches, in which the simulated drainage of each main ditch was the sum of drainage of its controlling HRUs.

We revised the name of subtitle 2.3.2 to "Parameterization of distributed RIWP model".

**Comment4:** Crop growth is closely with ET? What are the model simulation performances of cash crops growth (biomass, LAI, phonology) and grain yield in the calibration and validation systems in the section of 3.1.

**Response:** Yes. The crop ET module embedded in the regional RIWP model is based on FAO

Irrigation & Drainage 56 ($ET_m = K_c * ET_0; ET_0 = \frac{0.408\Delta(R_n - G) + 900\gamma u_2 \frac{(e_s - e_a)}{T+273}}{\Delta + \gamma(1 + 0.34 u_2)}$ ) and the equation developed by Pereira et al. (2007) ($\frac{ET_{a\,ws}}{ET_m} = K_{sc} = K_{ss}K_{sw} = \left[1 - \frac{b}{100 * k_y}(EC_e - $

$EC_{et})\right]\frac{TAW_{salt} - D_r}{(1 - p_{cor})TAW_{salt}}$) to estimate crop actual ET under water stress and/or saline condition.

Actual ET is affected by the soil water and salt content in the crop current root zone, and due to the crop root growth during the growing season the crop root zone is changing with time. We applied an empirical equation to quantify the crop root depth change with time in our ET module.

In one hand, ET is affected by the soil water and salt content in the root zone, on the other hand,

ET will affect the soil water and salt content in the root zone due to its role of water balance component. Thus, crop growth is closely connected to ET in our study. We did not include the estimation of biomass such as LAI, crop height in the ET and yield estimation module in our study. Also, as crop yield is actually affected by the crop actual ET during the growing season, we used the model of Stewart et al. (1977) ($\frac{Y_a}{Y_m} = \prod_{j=1}^{n=4}\left(1 - k_y\left(\frac{ET_{aj}}{ET_{mj}}\right)\right)$) to calculate crop yield in our study, in which crop ET and yield has a positive correlation. However, due to the lack of yield data, we only calibrated regional ET and made validation, and the model simulation indicated a reasonable performance of regional ET.

**Comment5:** Each section of the three Results and Discussion is needed for greater improvement especially in global sensitivity analysis and irrigation water productivity. Provide more explanations regarding the cause of simulation results, except for comparison with similar previous study results.

**Response:** Thanks very much for this comment and suggestion. We have made further explanations of the cause of the simulation results in each section of the three Results and

Discussion. In section 3.1 Model performance, we added "Besides, the cumulative $ET_{RS}$ was taken by the 8 times of daily ET on satellite acquisition date, thus using the non-representative ETRS above the average daily value may also result in the underestimation of $ET_{IWP}$." and "In the uncultivated area (Fig.7a), simulated groundwater table level presented a slower and more flat decreasing trend than measured value. By assuming a completely non-vegetation coverage condition of uncultivated area while it is not actually the case, estimated groundwater evapotranspiration driven by capillarity will become smaller than its actual value, in which small vegetation will transpires amounts of water from soil and soil moisture is relatively low thus groundwater evapotranspiration is higher." in L471-473 and L479-484 of revised manuscript. In section 3.2 Global sensitivity analysis, we added "Due to the high sensitivity of IWP, groundwater table depth and salinity to the specific yield, it is highly recommended to use spatially variable values of specific yield rather than a constant one as a model input if it is available, which could greatly enhance the evaluation accuracy of the RIWP model. Also, it is indicated that the permeability coefficient of unconfined aquifers (K) did not significantly affect the IWP, groundwater table depth and salinity. Due to the lack of measurement data in our study, we adopted a unified K value for the whole study area, which also make the model simulations reasonable for their insensitive to this parameter." in L509-515 of the revised manuscript. In section 3.3 Regional irrigation water productivity, we added "Note that these IWP values were based on the simulated water balance and crop yields of individual HRU, which may deviate to a certain extent from the real values. It can still represent the utilization of water resources at the regional scale." and "As we can see in Fig. 9, the simulated IWP values for three crops were lower in the south, west, north and north-west of the JFID than in the other regions. The south of the JFID is the main canal for water diversion, which provide higher irrigation quota than other regions, in which results in a lower IWP. For the west of JFID, it is mainly uncultivated area, thus the IWP is lower than other regions. In the north-west of the JFID, main drainage ditch received the drainage water with high saline content from four sub-main ditches and drained all the way to the north of JFID. Ditch seepage water with high salinity resulted in the severe soil salinization in the north and north-west of JFID, which will restrict the crop growth and lower the IWP." in L521-524 and L547-551 of the revised manuscript.

**Comment 6:** L705, what are the measured values? Detail on figure title.

**Response:** Sorry about not describing the parameter value ranges in Table 3. These are the possible parameter value ranges of this study area, which referred to the local measurements, survey data and relevant research papers. We revised the Table title to "Table 3. The collected possible parameter variation ranges and calibrated values of the parameters describing soil hydraulic characteristics ($K_e$, $S_y$, $K$) and irrigation and drainage system ($\eta_{lc}$, $\eta_{fc}$, $\gamma_d$, $A$, $m$)." in L828-830 of revised manuscript. We added a note below the Table 3 to explain the source of the possible parameter value ranges in L831-835 of the revised manuscript as following:

"Note: The parameter value ranges were collected from local measurements, survey data and relevant research results. Soil texture of canal bed was silty sandy loam for 0-1 and 2-3 m depth below the ground, and sandy loam for 1-2 m. For silty sandy loam soil, the bulk density and saturated soil water conductivity are 502.3 mm d$^{-1}$ and 1.42gcm$^{-3}$, respectively. For sandy loam soil, the bulk density and saturated soil water conductivity are 1.49g cm$^{-3}$ and 592.6 mm d$^{-1}$, respectively. There were fine sand and sandy soil in the phreatic layer." And corresponding adjustment was made to the table title in L785-787 of the revised manuscript.

**Comment7:** Each section of L704 provide details on soil particle size, bulk density, saturated water conductivity in table 3.

**Response:** Sorry about the unclear expression of the soil texture and its hydraulic characteristics in Table 3. We have provided details about the soil particle size, bulk density and saturated water conductivity for canal bed and the phreatic layer in the note below Table 3 in L832-835 of the revised manuscript as "Soil texture of canal bed was silty sandy loam for 0-1 and 2-3 m depth below the ground, and sandy loam for 1-2 m. For silty sandy loam soil, the bulk density and saturated soil water conductivity are 502.3 mm d-1 and 1.42gcm$^{-3}$, respectively. For sandy loam soil, the bulk density and saturated soil water conductivity are 1.49g cm$^{-3}$ and 592.6 mm d$^{-1}$, respectively. There were mainly fine sand and sandy soil in the phreatic layer."

**Comment8:** Figure 10, there was no obvious difference in irrigation water productivity in groundwater 0-1 and 1-2 m? If not, provide the corresponding results between these groundwater levels

**Response:** Thanks very much for this comment. Yes, there was no obvious difference in irrigation water productivity between groundwater table depth in the range of 0-1 and 1-2m. When groundwater table level is shallower (0-1m), more groundwater evapotranspiration could contribute to crop water use, which will increase the irrigation water productivity. On the other hand, due to the high groundwater salinity bigger soluble salt content will go into the crop root zone, which enhance the salt stress on crop water use and thus decrease the irrigation water productivity. Similar, deeper groundwater table level will contribute less groundwater evapotranspiration but also less salt content to root zone for crop water use. In this way, the irrigation water productivity under the 0-1 m groundwater table depth was not obviously different from that under the 1-2 m groundwater table depth.

**List of all relevant changes corresponding to the comments of Editor:**

[revised manuscript text omitted]

**Comment8:** No change in context

**Responses to the comments of Reviewer #2:**

**Recommendation:** I like this paper and believe it should be published with medium and minor edits. It is well-written and structured but will need some copy-editing as some of the English grammar and syntax can be improved. The main changes should relate to how the authors can make their model and its results more reader-friendly in that readers will want to know how this model helps users and managers better manage irrigation water. With the current version, it is not clear at the moment where these insights sit. In other words the author's own interpretation of their

RIWP model needs to be more clearly written.

**Response:** We are appreciating to the reviewer for the useful comments and suggestions to the paper. We have made corresponding changes to improve the English grammar and syntax to improve the quality of this paper. In the sections of abstract and conclusion, we added the context about explaining how this model could be used by different stakeholders in irrigation water management, which makes this paper much more reader-friendly. Below are the detailed responses to all comments. We cited first the comment, which is followed by our response and often by a section how the text will be revised in the manuscript. The text in blue are changes and additions in the original text. For clarity we do not show the removed text in the blue content.

**Substantive comments:**

**Comment1:** The productivity model depends on four parameters; water supply from irrigation open canals, field crop water consumption, groundwater drainage into open ditches, and groundwater lateral flow. Can the authors explain why rainfall is not included in their model as a water supply to crop growth? How would the model work in an area with more rainfall than in their case study?

**Response:** Thanks very much for this useful comment. We are sorry for not explaining clearly in the original context. Contribution of rainfall is actually included in the field scale irrigation water productivity module, which is a developed field IWP model to simulate field water, salt, ET and crop yield under shallow groundwater condition. Rainfall is considered as an input of the vertical water balance equation contributing to crop growth. Detailed context and equation about considering rainfall in the water balance equation in field scale IWP model, referred to Xue et al., (2018), are as following:

Daily water and salt balances are required for the estimation of daily $ET_a$. Water balance in current root zone is as following:

$$WCr_i = WCr_{i-1} + R_{i-1} + I_{i-1} + RG_{i-1} + Gwr_{i-1} - ETa_{i-1} - Pwr_{i-1}$$

Thus, this model is reasonable and applicable for an area with more rainfall than in our case study.

**Comment2:** Can the authors explain why lateral movement between drainage 'bonds' the units together (line 160) but that lateral movement of irrigation water down channels does not? Surely irrigation water and drainage water are both moving laterally as well as vertically?

**Response:** Thanks very much for this useful comment. Irrigation water and drainage water are surely moving laterally and vertically. We are sorry about not explaining clearly in the original text. We are talking about the lateral exchange between adjacent groundwater units here, not the lateral water movement caused by drainage or irrigation conveyance. The study area is the arid region with shallow groundwater, which can be a very important water contribution source to crop growth. Due to the seepage loss from unsaturated soil profile to shallow groundwater and groundwater evapotranspiration going upward to unsaturated soil profile, the phreatic layer will be unstable and the groundwater table level will vary with it. Based on daily time step, we assumed that the groundwater level is unified in each HRU and the process of lateral water exchange of the phreatic layer between two adjacent HRUs were completed within one day. Additionally, it is indicated that the main irrigation canals and drainage ditches directly connect with groundwater and can be considered as the side boundaries in the model in lines 153-154 of original context.

**Comment3:** Seepage loss from channels is in the model, but I do not readily spot where seepage loss beneath the root zone from fields is accommodated?

**Response:** Thanks very much for this useful comment and suggestion. Sorry for not explaining clearly in the manuscript. Just like mentioned in comment1 that contribution of rainfall to crop growth is not readily spotted, the seepage loss beneath the root zone from fields is also included in the former developed field scale irrigation water productivity module. Seepage from crop root zone to deeper soil profile like potential root zone (Pwr), transmission zone and phreatic layer (Pwg) are considered as the components of water balance equation in the vertical soil profile. Detailed context and equation about considering field scale irrigation seepage in the water balance equation in field scale IWP model, referred to Xue et al., (2018), are as following:

Water balance in current root zone

$$WCr_i = WCr_{i-1} + R_{i-1} + I_{i-1} + RG_{i-1} + Gwr_{i-1} - ETa_{i-1} - Pwr_{i-1}$$

Water balance in potential root zone

$$WCg_i = WCg_{i-1}+Pwr_{i-1}-RG_{i-1} - Gwr_{i-1} - Pwg_{i-1} + Gwg_{i-1}$$

Groundwater balance

$$hg_i = hg_{i-1} - (1/S_y)(Pwg_{i-1} - Gwg_{i-1} - ext_{i-1})$$

**Comment4:** The authors write on page 17 a statement that the contribution of groundwater and proportion of non-beneficial soil evaporation are major influences on water productivity of their chosen crops. This seems to indicate that the productivity model is simply a biomass model related to the proportion of total water supply that ends up in transpiration? But there are other factors such as irrigation timing and scheduling that affect productivity. This makes this reviewer wonder what are the units of RIWP? And why are these units not utilised frequently throughout the paper?

Thus in other words is this a production model not a productivity model?

**Response:** Thanks for this useful comments. We explained in the first paragraph of the

Introduction in the original paper that IWP is defined as the crop yield per cubic meter of irrigation water supplied, and the unit of IWP is kg/m$^3$. The model is based on field ET of crop muti-growth stages and ET is computed with field daily hydrological model driven by irrigation scheduling, precipitation events, meteorology, and groundwater levels dynamics. As a result, irrigation scheduling has significant impact to field daily ET of different crop growth stages and final IWP. Furthermore, RIWP is the spatial distribution of IWP for an irrigated area, which is likely a map of IWP for different crops at the regional scale. Our RIWP model simulates yield response to water of different crops at the regional scale and is particularly suited to address conditions where water is a key limiting factor in crop production. It also provides an indicator which assesses the performance of the system, through the IWP or the yield that is produced per unit of irrigation water applied. Thus, we believe our model is more like a crop water productivity model.

**Comment5:** Also can the authors explain why, if nearly all the groundwater supplies and movements of water which are in the model come from irrigation both in the short and long-term, and not from rainfall or wider hydrogeological inflows, does the model 'bother' with groundwater as a factor determining IWP? Surely the main determinant of irrigation productivity in an entirely arid region is really only 'irrigation supply'. This reviewer knows partly the answer but the authors must not assume the readers know this distinction.

**Response:** Thanks very much for this comment and suggestion. We are sorry for not considering the reader-friendly part for this paper. In arid region with shallow groundwater, irrigation caused seepage goes into groundwater and is stored in there temporarily. It looks like that the irrigation seepage is not consumed by crop and is counted in the non-beneficial irrigation water use. However, groundwater evapotranspiration will also go upward and contribute to crop water use, which makes the irrigation seepage water reusing by crop come true. This will increase the beneficial use of irrigation water and thus improve the IWP. Therefore, groundwater is also an important factor determining IWP in arid region with shallow groundwater. We have made further explanations of reason in L63-69 in the revised manuscript as following:

"Furthermore, by changing hydrological processes, irrigation and drainage affect water and salt dynamics in crop root zone, groundwater, and, eventually, crop production (Morison et al., 2008; Bouman et al., 2007). Specifically, in arid region, irrigation-caused deep seepage is the mainly recharge of groundwater. Shallow groundwater can in turn go upward and contribute to crop water use by capillary action, which means the irrigation seepage can be reused by the crop growth to improve IWP. Thus, RIWP analysis requires the quantification of the complex agro-hydrological processes, including soil water and salt dynamics, groundwater movement, crop water use and crop production."

**Comment6:** Line 490 – can the authors explain why productivity declines when water supply from irrigation goes up? This may be consistent with other results, but it is counter to expectation? (Again the problem is that the units of IWP are not given in the main body of the paper).

**Response:** Sorry about not describing the definition and unit of IWP clearly in the main text of this paper. We make corresponding revision in L61-62 of revised manuscript as following:

"IWP is defined as the crop yield per cubic meter of irrigation water supplied, and the unit of IWP is kg/m$^3$ (Singh et al., 2004)."

Water productivity declines when water supply from irrigation goes up. This is because of the shallow groundwater condition of our case study. Irrigation water amount directly affects soil moisture of crop root zone and finally decides the crop yield. As is well-known, crop yield is directly linked to actual ET. Decreasing irrigation water depth results in a reduction of actual ET, while actual ET decreases slower than irrigation water depth does because of the contribution of groundwater evapotranspiration to crop water use (actual ET), which is directly linked to crop yield.

Thus, as the ratio of crop yield and irrigation water amount, irrigation water productivity increases when irrigation water amount decrease.

**Comment7:** Can the authors be clear about what m3 of water on the denominator is about – is it total supply in cubic meters or is it total transpired cubic meters?

**Response:** Thanks for this useful comment. The $m^3$ of water on the denominator is the total supply in cubic meters. Also, as it is indicated in section 2.2 that the field irrigation water amount is the input of the field IWP module, which generates the IWP results for three crops in each HRU

and map the spatial distribution of RIWP. We also revised the statement in L61-62 of revised manuscript as following:

"IWP is defined as the crop yield per cubic meter of irrigation water supplied, and the unit of IWP

is $kg/m^3$ (Singh et al., 2004)."

**Comment8:** As a key comment, I think Section 3 needs to be re-written by starting or leading with key management results and insights that are readable by different stakeholders. At the moment this section is written with the model rather than the results in mind. The key management insights are buried deep within this section and are not easy to find. Here are some guide questions that show what I mean:

Which affects crop productivity more – irrigation dose/depth applied or the contribution from groundwater?

Which affects crop productivity more – lots of shallow irrigation applications or fewer deeper applications?

Which type of crop is most productive in coping with water supply coming from non-irrigation sources?

How is productivity negatively or positively affected by a combination of drainage and salts?

What explains the changing 'red spots' of high productivity in the maps in Figure 9 and whether and how this high level of productivity can be extended to the rest of the Jiefangzha Irrigation

District so that everything becomes 'red'.

I hope these examples show why the 'results' section currently does not clearly guide managers and planners.

**Response:** Thanks very much for these useful comments. As results of a new developed model, we firstly describe the performance of the model, followed by the parameters sensitivity analysis.

At last, we try to get some insight of RIWP with the model. We revised the expression of model results to make them more reader-friendly to different stakeholders according to this reviewer's suggestion. Finally, these parts are arranged in the revised manuscript with following sequence:

L442-445: Good agreements were obtained by RIWP model in simulating IWP and hydrological components during the calibration and validation periods. Table 2 tabulated the calibrated parameters describing crop growth and water usage, and Table 3 tabulated the possible variation ranges and calibrated values of the parameters describing soil hydraulic characteristics and irrigation and drainage system.

L495-501: We concluded that for shallow groundwater buried area like JFID, sometimes the effect of groundwater contribution on IWP would be greater than that of irrigation water depth applied.

Applying lots of shallow irrigation to the crops may reduce the deep percolation and decrease the non-beneficial water use in evaporation. Applying fewer and deeper irrigation water applied will result in deeper percolation meanwhile greater groundwater contribution to beneficial crop water use. Thus, compared with lots of shallow irrigation applied, applying fewer deeper irrigation schedule may have greater effect on IWP in arid regions with shallow groundwater.

L524-532: We could see there are "red HRUs" in Figure 9 changing with time and space due to different irrigation water depth applied under different groundwater conditions. Even different crop species can result in big difference in IWP…. This was because that the irrigation quota was reduced over this period, and the contribution of groundwater compensated the crop yield losses. With less irrigation water applied, the number of "red HRUs" will increase along with it.

L541-557: Particularly, when the farmlands had limited supply of irrigation water, the groundwater table depth and salinity played an important role on IWP. Through the drainage ditches, groundwater could drain both water and salt out of the field, thus the groundwater table level declines and the soluble salt content going upward along with groundwater evapotranspiration to crop root zone decreases. Despite the negative effect of draining water on IWP, the positive effect of draining salt out of the field will positively affect IWP….. Thus, properly groundwater drainage management and dealing with salt accumulation at the end of main drainage ditches in an irrigated area is also a pressing and unsolved problem for increasing the "red HRUs", which needs to be figured out by irrigation managers.

L558-561: As the major food-producing region of China, improving water productivity means producing greater amounts of food crops with less amount of water, based on local or regional potential. With declining access to water resources, farmers will need to grow different crops to maintain or increase crop production profitability in the future.

L566-568: Thus, planting sunflowers should be promoted in the JFID when available irrigation water resources is declining in the future, and this practice will definitely increase the "red HRUs".

**Comment9:** Can the authors also introduce some 'future or methodological critical thinking'? In other words, how does such an approach really guide current managers in improving irrigation management? What future improvements to the method and model might allow this to happen? How does the author's model differ from other regional irrigation productivity studies, eg. conducted by the Water for Food Institute, Nebraska.

**Response:** Thanks very much for this comment. Other regional irrigation productivity models, such as Aqua crop, consider the crop yield response to water and temperature stress. It also simulates soil evaporation and crop transpiration explicitly as individual processes. Aqua crop simulates the growth, biomass production, and harvestable yield. It did not take fully consideration of groundwater on crop water use and production. Differently, our RIWP consider the regional hydrological processes including water and salt stress on crop yield and IWP, and soil evaporation and crop transpiration processes are simulated together as evapotranspiration in this model.

Because that IWP is the final and most important simulation index in RIWP model, only crop yield is simulated in our model while the crop biomass part are not included. The groundwater module in RIWP model can also capture the effect of shallow buried groundwater level and salinity on crop water use, which is very common in arid region with shallow groundwater. We added some future thinking and suggestions to irrigation managers in improving irrigation management based on our developed model in results and conclusion section of revised manuscript:

L558-561: As the major food-producing region of China, improving water productivity in JFID

means producing greater amounts of food crops with less amount of water, based on local or regional potential. With declining access to water resources, farmers will need to grow different crops to maintain or increase crop production profitability in the future...........Thus, planting sunflowers should be promoted in the JFID when available irrigation water resources is declining in the future.

L591-598: Thus, keeping the groundwater table depth in the optimal range and sustainable is of great importance to reach higher crop IWP at the regional scale, irrigation managers may need to reasonably determine the irrigation quota and constantly maintain the drainage system.

Groundwater sustainability includes spacing withdrawals to avoid excessive depletion and taking measures to safeguard or improve groundwater quality. To achieve this, regional irrigation managers may need to take monitoring efforts to establish historic and current conditions, research to model groundwater systems, forecast future variation, and policy to manage activities influencing groundwater table and quality.

L616-627: Programmed in Matlab (Mathworks Inc., 2015), RIWP model can be run on different operating systems. Furthermore, the model includes capability for parallelization of simulations to reduce batch run times when conducting simulations over large areas, conditions, and/or time periods. In the nearly future, enabling the code to be linked quickly with other disciplinary models to support integrated water resource management could be a great improvement of RIWP model. Also, we are going to develop a website used for long-term distribution of the RIWP model and associated documentation. Finally, RIWP model could improve knowledge of best practices to enhance water productivity for key irrigation decision-makers. The simplicity of RIWP model in its required minimum input data, which are readily available or can easily be collected, makes it user-friendly. It is also a very useful model for scenario simulations and for planning purposes, which can be used by economists, water administrators and managers working in the arid irrigated area with shallow groundwater.

**Minor comments:**

**Comment1:** Be consistent "water productivity model" in title, but "water productivity estimation" in key words.

**Response:** Sorry for not being consistent through the context. We revised the "water productivity estimation" to "water productivity model" in key words of revised manuscript.

**Comment2:** Is there a substantive difference between "irrigation water productivity (IWP)" and "regional irrigation water productivity (RIWP)"

**Response:** Thanks very much for this comment. Yes, irrigation water productivity is a definition, which is the crop yield per cubic meter of irrigation water amount. Regional irrigation water productivity represents the spatial distribution of irrigation water productivity, which is much more like a map of irrigation water productivity at the regional scale.

**Comment3:** Line 36. Are uncultivated lands bare lands, or natural vegetation?

**Response:** Thanks very much for this comment. The uncultivated lands, merely bare soil, accounted for about 34% of our study area. We explained this in line 435-436 of original manuscript as: The uncultivated area, merely bare soil, accounted for about 34% of the JFID, and the $ET_{IWP}$ of uncultivated area was merely soil evaporation. To avoid misleading readers in the former context, we corrected the expression of the sentence in L35-36 of revised manuscript as following: In each HRU, we considered four land-use types: sunflower fields, wheat fields, maize fields and uncultivated lands (merely bare soil).

**Comment4:** Line 45. I would use the words 'depth applied' or 'delta and deltas' when discussing water applied via irrigation (and not 'depth' alone). Otherwise this use is confusing "when groundwater table depth is in the range of 2 m to 4 m, regardless of irrigation water depths"

**Response:** Thanks very much for this comment. We made corresponding revisions in the context. All the "irrigation water depth" in the manuscript were rewritten to "irrigation water depth applied".

**Comment5:** Line 54. I would not use a single figure of 90% here "where irrigated agriculture accounts for about 90% of the total". I would use a range e.g. 70 to 90%

**Response:** Thanks very much for this comment. We revised the number to 70 to 90% and added the reference in the L56-58 of revised manuscript as following:

Especially, in arid and semi-arid regions of the world, where irrigated agriculture accounts for about 70 to 90% of the total water use (Jiang et al., 2015; Gao et al., 2017, Dubois, 2011)…

**Comment6:** Line 69 Field experiments may be costly but they do allow for calibration and an understanding of the relevant parameters and processes "but field experiments are expensive and time consuming, making it unsuitable for regional evaluation of IWP." So field experiments still help with a regional evaluation?

**Response:** Thanks very much for this comment. Just like the reviewer said that field experiments may be costly but then do allow for calibration and understanding of the relevant parameters and processes. We are able to adopt the field experiment to accurately evaluate the IWP at the field scale. For a larger scale such as a watershed or an irrigated area, using field experiment to evaluate the IWP of multiple spots within the area of interest may not be a good way to reproduce the spatial distribution of IWP for its time-money consuming and lack of basic regional hydrological processes. However, after we obtain the evaluation results for regional hydrological processes and IWP, field experiments can still be helpful with the calibration part.

**Comment7:** Line 84, can an example of simplified distributed models be given? "There are two types of distributed hydrologic models that are used to integrate with crop models: numerical distributed models, such as SWAT and MODFLOW, and simplified distributed models based on water balance equations."

**Response:** Thanks very much for this comment. We are sorry about not explaining it clearly. We gave two example of simplified distributed models called FARME and HEC-HMS in L87-97 of the revised context as following:

"There are two types of distributed hydrologic models that are used to monitor complex regional hydrological processes: numerical distributed models, such as SWAT and MODFLOW, and simplified distributed models, such as FARME (Kumar and Singh, 2003) and HEC-HMS (USACE,

1999) based on water balance equations. Numerical, process-based models consider the entire complexity and heterogeneity of regional hydrological systems. MODFLOW is commonly used for groundwater dynamics simulation (Kim et al., 2008). But it is limited in well-monitored large irrigation areas, due to the large number of parameters and input data required. SWAT is used to simulate land surface hydrologic and crop growth processes. It relies on the digital elevation model (DEM) to delineate surface water flow pathways. However, many irrigation areas are quite flat, and surface water flow pathways are controlled by irrigation and drainage systems, instead of terrain elevation differences. Furthermore, SWAT alone does not describe the complex interactions between groundwater and soil water, which are fundamental in arid and semi-arid areas with shallow groundwater."

**Comment8:** Line 94 – suggest small change "However, the large spatial grids can hardly reflect the regional complex cropping pattern heterogeneity, and the large temporal steps cannot capture daily soil water" to this "However, the large spatial grids poorly reflect the regional complex cropping pattern heterogeneity, and the large temporal steps cannot capture daily soil water"

SWAT alone does not describe the complex interactions between groundwater and soil water, which are fundamental in arid and semi-arid areas with shallow groundwater".

**Response:** Thanks very much for this comment. We have revised the original sentence to the recommended one in L100-104 of revised manuscript as following:

"However, the large spatial grids poorly reflect the regional complex cropping pattern heterogeneity, and the large temporal steps cannot capture daily soil water and salt dynamics which is essential for crop growth simulation. SWAT alone does not describe the complex interactions between groundwater and soil water, which are fundamental in arid and semi-arid areas with shallow groundwater."

**Comment9:** Line 139 The authors could do better in explaining what an HRU is? Is it an abstract artefact, or a real command unit within an irrigated landscape? Do irrigation managers use HRUs?

**Response:** Sorry for not explaining HRU more specifically. The hydrologic response unit (HRU) is an abstract artefact created by model developer, which provides an efficient way to discretize large watersheds where simulation at the field scale may not be computationally feasible. For a regional area, the smallest spatial unit of its hydrological processes is not generally defined by physically meaningful boundaries. The HRU is like the smallest spatial unit of the model, and the standard HRU definition approach lumps all similar land uses, soils, and slopes within a sub-basin based upon user-defined thresholds. HRU is more widely used by regional hydrological model developers and users, which may include some of the irrigation managers or researchers. Following are the revised context in L151-155 of the revised paper:

"The HRU is an abstract artefact created by hydrological developer and is like the smallest spatial unit of the model, which provides an efficient way to discretize large watersheds where simulation at the field scale may not be computationally feasible. In each HRU, soil texture and groundwater conditions are assumed to be homogeneous, but different cropping patterns can exist."

**Comment10:** Line 230 can this sentence about boundaries be explained? "There are three types of groundwater boundaries: river boundaries, drainage ditch boundaries and no flux boundaries"

**Response:** Thanks very much for this comment. Sorry for not explaining the boundary types specifically in the original paper. We revised the context in L241-246 of revised manuscript as following:

"There are three types of groundwater boundary conditions: river head (when the boundary HRU including irrigation canal and the daily river flux equals to the daily canal flux), river flux (when the boundary HRU including drainage ditches and the water heads in ditches are assumed constant and equal to the river head) and constant flux (when the boundary HRU is mainly barren area and no irrigation is applied, thus in our study 0 flux is assumed)."

**Comment11:** Line 258 spelling/grammar? "Cropping patterns are complex for each HRU and sometimes HRU include uncultivated land, forest". This should be "Cropping patterns are complex for each HRU and sometimes HRUs include uncultivated land, forest"

**Response:** Thanks very much for this comment. We revised the sentence to "Cropping patterns are complex for each HRU and sometimes HRUs include uncultivated land, forest" in L274-275

of the revised manuscript.

**Comment12:** Line 293 – correct this sentence to "Considering the high spatial heterogeneity, meteorological data need to be collected from all the weather stations within or close to the study area."

**Response:** Thanks very much for this comment. We made corresponding revision in L309-310 of the revised paper as following:

"Considering the high spatial heterogeneity, meteorological data need to be collected from all the weather stations within or close to the study area."

**Comment13:** Line 427 check grammar to this "the ditches of the same order share the same the drainage coefficient, assuming well-operated conditions. However,"

**Response:** Thanks very much for this comment. We are sorry for not express it clearly and made corresponding correction in L459-460 of the revised context as following:

"In the model, for each year, we adopt same drainage coefficient for all the ditches of the different orders, assuming a well operated condition."

**Comment14:** Line 502 – difficult to follow the argument with the current English. Should this not read "indicates that when irrigation applied decreased from 300<IWD<400mm to

200<IWD<300mm it lead to decreases in IWP caused by a reduction of ET." (But this seems to contradict statements made elsewhere in the paper?

**Response:** Thanks very much for this comment. Sorry for not expressing the result clearly. We made corresponding correction in L584-586 of the revised paper as following:

"…and it indicates that when irrigation applied decreased from 300<IWD<400mm to

200<IWD<300mm it leads to decreases in IWP, which is caused by faster reduction of ET than irrigation applied."

For the potential reason of this result, we made further explanation in the following sentences in the original paper. Due to the shallow buried groundwater table condition, groundwater contribution will make up for ET reduction when we applied smaller irrigation water amount. As most of the IWP variation rules under sallow groundwater condition in this paper, when the speed of reduction of irrigation water applied is higher than the reduction of ET, IWP increases.

However, when irrigation water applied decreases from 300<IWD<400mm to 200<IWD<300mm at this time, IWP decreases, which means that there exists another reason accelerate the reduction of ET. Thus, we deduced that in this situation less irrigation water will weaken the role of irrigation on salt leaching and result in more severe salinization in crop root zone. The negative effect of salt stress on crop water use is greater than the positive effect of shallow groundwater contribution on crop water use at this situation.

**Comment15:** Line 505 onwards – very difficult to understand this text! "ET, which is less irrigation water will weaken the role of irrigation on salt leaching and result in more severe salinization in crop root zone. Thus, reasonably determining the irrigation quota and constantly maintaining the drainage system to keep the groundwater table depth in the optimal range is of great importance to reach higher crop IWP at the regional scale."

**Response:** Sorry for not clearly expressing the result and reason of it. We made corresponding correction in L586-594 of revised manuscript to make it easier to read as following:

"Shallow buried groundwater contribution will make up for ET reduction when smaller irrigation water applied, thus there exists another reason accelerate the reduction of ET. We deduced that less irrigation water would weaken the role of irrigation on salt leaching and result in more severe salinization in crop root zone. The negative effect of salt stress on crop water use is greater than the positive effect of shallow groundwater contribution on crop water use at this situation. Thus, keeping the groundwater table depth in the optimal range is of great importance to reach higher crop IWP at the regional scale, irrigation managers may need to reasonably determine the irrigation quota and constantly maintain the drainage system."

**Comment16:** Line 511. Does not make sense "In view of the particularity of irrigated areas, taking fully consideration of the supply," Perhaps this? "In view of the heterogeneous conditions of irrigated areas, taking fully consideration of the supply,"

**Response:** Thanks very much for this comment. We revised the original sentence following your recommendation in L600 of revised manuscript as:

"In view of the heterogeneous conditions of irrigated areas, taking fully consideration of the supply…"

**List of all relevant changes corresponding to the comments of Editor:**

**Substantive Comment1:** No change in context

**Comment2:** No change in context

**Comment3:** No change in context

**Comment4:** L119 "irrigation water depth applied"

L213 "irrigation water depth applied"

L525 "irrigation water depth applied"

L569 "irrigation water depth applied"

[revised manuscript text omitted]

---

## Editor Decision (ED1)

A novel regional irrigation water productivity model for complex cropping patterns in arid regions coupling soil water and salinity dynamics, irrigation and drainage, and shallow groundwater movement

The manuscript has been reviewed by two reviewers, and the authors have, in my view, responded adequately to the issues raised.

I have, however, two additional issues that were not raised by the reviewers, and which I like to share with the authors. I invite the authors to take my comments into account when submitting an improved version of the paper.

1. There are some references in the text that do not appear in the reference list; take for example the references cited in lines 55 to 62: of the 9 references, 5 do not appear in the reference list. Also check the reference in line 174.

2. The manuscript is inconsistent with its units. All water fluxes should have a time dimension. So $W_{ls}$ (line 193-194) is the groundwater recharge per unit; and in your model you use a daily time step, so the correct unit is m/day. Same for $W_{as}$ (line 202-203), $I_n$ (line 203), $D_g$ (line 213), $W_{gr}$ (line 224), $P_{wg}$ (lines 251-252), $G_{wg}$ (lines 252-253). Check the correct unit of K (permeability coefficient, lines 224-225), I think it should have a time dimension. Check eq. 10 on consistency of the units/dimensions.

3. The amount of irrigation water applied seems small (lines 320-323); I calculated an average gross irrigation application of 162 mm/year [$(12 \times 10^8)/(0.66 \times 1.12 \times 10^6 \times 10^4)=0.162$ m/year]. Kindly explain.

4. Lines 388-391: What are thresholds for acceptable and good model performance for the 3 evaluation criteria used (NSE, R2 and RMSE)?

5. Line 451: "readily available groundwater"; here I think you deal with the unsaturated zone, so do you rather mean: "readily available soil moisture"?

6. Figure 9: in an earlier iteration I asked the authors to improve the colour-scheme of this figure. You have done so, but in the process, you have, unfortunately, not standardized the scales (as you had done in the original version of this figure, and as you have also correctly done in your figure S3). For the reader it is therefore very difficult to compare the different years. So for each crop redraw the maps by keeping the colour scale fixed over the years.

7. Figure 9 once more: none of the maps contain blank pixels – this suggest that each pixel in all years have values for the productivity of all three crops. This I find highly surprising, and in fact unlikely, (but I admit that I do not know the irrigation district). Please explain.

8. Section 3.2.1 concludes about which crops have the highest productivity (lines 481-486). Here productivity in money value (expressed e.g. in US\$/m³ or RMB/m³) would be the most

convincing criterion. Do you have average farm gate prices of the three crops, so that you can convert the IWP (kg/m$^3$) into RMB/m$^3$? You suggest that sunflower has a much higher "benefit" (line 485) than wheat. Do you mean "price"?

9. The manuscript still is weak in grammar, and reviewer #2 did a great job to highlight the major weaknesses. Please also check the following lines: 15, 72, 98, 146, 196, 206, 269, 293, 295, 296, 315, 364, 427.

I encourage the authors to submit a revised version of the paper, taking also the above details into account.